# Low-distortion *and* GPU-compatible Tree Embeddings in Hyperbolic Space

**Max van Spengler** [1]   **Pascal Mettes** [1]

## Abstract

Embedding tree-like data, from hierarchies to ontologies and taxonomies, forms a well-studied problem for representing knowledge across many domains. Hyperbolic geometry provides a natural solution for embedding trees, with vastly superior performance over Euclidean embeddings. Recent literature has shown that hyperbolic tree embeddings can even be placed on top of neural networks for hierarchical knowledge integration in deep learning settings. For all applications, a faithful embedding of trees is needed, with combinatorial constructions emerging as the most effective direction. This paper identifies and solves two key limitations of existing works. First, the combinatorial construction hinges on finding highly separated points on a hypersphere, a notoriously difficult problem. Current approaches achieve poor separation, degrading the quality of the corresponding hyperbolic embedding. We propose highly separated Delaunay tree embeddings (HS-DTE), which integrates angular separation in a generalized formulation of Delaunay embeddings, leading to lower embedding distortion. Second, low-distortion requires additional precision. The current approach for increasing precision is to use multiple precision arithmetic, which renders the embeddings useless on GPUs in deep learning settings. We reformulate the combinatorial construction using floating point expansion arithmetic, leading to superior embedding quality while retaining utility on accelerated hardware.

## 1. Introduction

Tree-like structures such as hierarchies are key for knowledge representation, from biological taxonomies (Padial et al., 2010) and phylogenetics (Kapli et al., 2020) to natural language (Miller, 1995; Tifrea et al., 2018; Yang et al., 2016), social networks (Freeman, 2004), visual understanding (Desai et al., 2023; Van Spengler et al., 2023; Pal et al., 2025) and more. To obtain faithful embeddings, Euclidean space is ill-equiped; even simple trees lead to high distortion (Sonthalia & Gilbert, 2020). On the other hand, the exponential nature of hyperbolic space makes it a natural geometry for embedding trees (Nickel & Kiela, 2018). This insight has led to rapid advances in hyperbolic learning, with superior embedding (Sala et al., 2018) and clustering (Chami et al., 2020) of tree-like data.

Recent literature has shown that hyperbolic tree embeddings are not only useful on their own, they also form powerful target embeddings on top of deep networks to unlock hierarchical representation learning (Peng et al., 2021; Mettes et al., 2024). Deep learning with hyperbolic tree embeddings has made it possible to effectively perform action recognition (Long et al., 2020), knowledge graph completion (Kolyvakis et al., 2020), hypernymy detection (Tifrea et al., 2018) and many other tasks in hyperbolic space. These early adoptions of hyperbolic embeddings have shown a glimpse of the powerful improvements that hierarchically aligned representations can bring to deep learning.

The rapid advances in hyperbolic deep learning underline the need for hyperbolic tree embeddings compatible with GPU accelerated software. Current tree embedding algorithms can roughly be divided in two categories; optimization-based and constructive methods. The optimization-based methods, e.g. Poincaré embeddings (Nickel & Kiela, 2017), hyperbolic entailment cones (Ganea et al., 2018), and distortion optimization (Yu et al., 2022b), train embeddings using some objective function based on the tree. While these approaches are flexible due to minimal assumptions, the optimization can be unstable, slow and result in heavily distorted embeddings. Conversely, constructive methods traverse a tree once, placing the children of each node on a hypersphere around the node's embedding (Sarkar, 2011; Sala et al., 2018). These methods are fast, require no hyperparameter tuning and have great error guarantees. However, they rely on hyperspherical separation, a notoriously difficult problem (Saff & Kuijlaars, 1997), and on multiple precision floating point arithmetic, which is incompatible with GPUs and other accelerated hardware.

[1]VIS Lab, University of Amsterdam, The Netherlands. Correspondence to: Max van Spengler <m.w.f.vanspengler@uva.nl>.

*Proceedings of the 42nd International Conference on Machine Learning*, Vancouver, Canada. PMLR 267, 2025. Copyright 2025 by the author(s).

Our goal is to embed trees in hyperbolic space with minimal distortion yet with the ability to operate on accelerated GPU hardware even when using higher precision. We do so in two steps. First, we outline HS-DTE, a new generalization of Delaunay tree embeddings (Sarkar, 2011) to arbitrary dimensionality through hyperspherical separation. Second, we propose HypFPE, a floating point expansion arithmetic approach to enhance our constructive hyperbolic tree embeddings. We develop new routines for computing hyperbolic distances on floating point expansions and outline how to use these on hyperbolic embeddings. Furthermore, we provide theoretical results demonstrating the effectiveness of these floating point expansion routines. Floating point expansions allow for higher precision similar to multiple precision arithmetic. However, our routines can be implemented using standard floating point operations, making these compatible with GPUs. Experiments demonstrate that HS-DTE generates higher fidelity embeddings than other hyperbolic tree embeddings and that HypFPE further increases the embedding quality for HS-DTE and other methods. The code will be made available at https://github.com/maxvanspengler/hyperbolic_tree_embeddings.

## 2. Preliminaries and Related Work

### 2.1. Hyperbolic Geometry Preliminaries

To help explain existing constructive hyperbolic embedding algorithms and our proposed approach, we outline the most important hyperbolic functions here. For a more thorough overview, we refer to (Cannon et al., 1997; Anderson, 2005). Akin to (Nickel & Kiela, 2017; Ganea et al., 2018; Sala et al., 2018), we focus on the Poincaré ball model of hyperbolic space. For $n$-dimensional hyperbolic space, the Poincaré ball model is defined as the Riemannian manifold $(\mathbb{D}^n, \mathfrak{g}^n)$, where the manifold and Riemannian metric are defined as

$$\mathbb{D}^n = \left\{ \mathbf{x} \in \mathbb{R}^n : ||\mathbf{x}||^2 < 1 \right\},$$
$$\mathfrak{g}^n = \lambda_{\mathbf{x}} I_n, \quad \lambda_{\mathbf{x}} = \frac{2}{1 - ||x||^2}. \quad (1)$$

Using this model of hyperbolic space, we can compute distances between $\mathbf{x}, \mathbf{y} \in \mathbb{D}^n$ either as

$$d_{\mathbb{D}}(\mathbf{x}, \mathbf{y}) = \cosh^{-1}\left( 1 + 2\frac{||\mathbf{x} - \mathbf{y}||^2}{(1 - ||\mathbf{x}||^2)(1 - ||\mathbf{y}||^2)} \right), \quad (2)$$

or as

$$d_{\mathbb{D}}(\mathbf{x}, \mathbf{y}) = 2\tanh^{-1}\left( || - \mathbf{x} \oplus \mathbf{y}|| \right), \quad (3)$$

where

$$\mathbf{x} \oplus \mathbf{y} = \frac{(1 + 2\langle \mathbf{x}, \mathbf{y} \rangle + ||\mathbf{y}||^2)\mathbf{x} + (1 - ||\mathbf{x}||^2)\mathbf{y}}{1 + 2\langle \mathbf{x}, \mathbf{y} \rangle + ||\mathbf{x}||^2||\mathbf{y}||^2}, \quad (4)$$

is the Möbius addition operation. These formulations are theoretically equivalent, but suffer from different numerical

errors. This distance represents the length of the straight line or geodesic between $\mathbf{x}$ and $\mathbf{y}$ with respect to the Riemannian metric $\mathfrak{g}^n$. Geodesics of the Poincaré ball are Euclidean straight lines through the origin and circular arcs perpendicular to the boundary of the ball.

We will use some isometries of hyperbolic space, as these conveniently allow for moving embeddings around without changing pairwise distances, keeping the distortion unaffected. More specifically, we will use reflections in geodesic hyperplanes. A geodesic hyperplane is an $(n-1)$-dimensional manifold consisting of all geodesics through some point $\mathbf{x} \in \mathbb{D}^n$ which are orthogonal to a normal geodesic through $\mathbf{x}$ or, equivalently, orthogonal to some normal tangent vector $\mathbf{v} \in \mathcal{T}_{\mathbf{x}}\mathbb{D}^n$. For the Poincaré ball these are the Euclidean hyperplanes through the origin and the $(n-1)$-dimensional hyperspherical caps which are perpendicular to the boundary of the ball. We will denote a geodesic hyperplane by $H_{\mathbf{x}, \mathbf{v}}$. Reflection in a geodesic hyperplane $H_{\mathbf{0}, \mathbf{v}}$ through the origin can be defined as in Euclidean space, so as a Householder transformation

$$R_{H_{\mathbf{0}, \mathbf{v}}}(\mathbf{y}) = (I_n - 2\mathbf{v}\mathbf{v}^T)\mathbf{y}, \quad (5)$$

where $||\mathbf{v}|| = 1$. Reflection in the other type of geodesic hyperplane is a spherical inversion:

$$R_{H_{\mathbf{x}, \mathbf{v}}}(\mathbf{y}) = \mathbf{m} + \frac{r^2}{||\mathbf{y} - \mathbf{m}||^2}(\mathbf{y} - \mathbf{m}), \quad (6)$$

with $\mathbf{m} \in \mathbb{R}^n$, $r > 0$ the center and radius of the hypersphere containing the geodesic hyperplane. We will denote a reflection mapping some point $\mathbf{x} \in \mathbb{D}^n$ to another point $\mathbf{y} \in \mathbb{D}^n$ by $R_{\mathbf{x} \to \mathbf{y}}$. The specific formulations and derivations of the reflections that we use are in Appendix A.

### 2.2. Related Work

**Hyperbolic tree embedding algorithms.** Existing embedding methods can be divided into two categories: optimization-based methods and constructive methods. The optimization methods typically use the tree to define some loss function and use a stochastic optimization method such as SGD to directly optimize the embedding of each node, e.g. Poincaré embeddings (Nickel & Kiela, 2017), hyperbolic entailment cones (Ganea et al., 2018) and distortion optimization (Sala et al., 2018; Yu et al., 2022b). Poincaré embeddings use a contrastive loss where related nodes are pulled together and unrelated nodes are pushed apart. Hyperbolic entailment cones attach an outwards radiating cone to each node embedding and define a loss that forces children of nodes into the cone of their parent. Distortion optimization directly optimizes for a distortion loss to embed node pairs. Such approaches are flexible, but do not lead to arbitrarily low distortion and optimization is slow. Constructive methods are either combinatorial methods (Sarkar, 2011;

Sala et al., 2018) or eigendecomposition methods (Sala et al., 2018). Combinatorial methods first place the root of a tree at the origin of the hyperbolic space and then traverse down the tree, iteratively placing nodes as uniformly as possible on a hypersphere around their parent. (Sarkar, 2011) proposes a 2-dimensional approach, where the points have to be separated on a circle; a trivial task. For higher dimensions, (Sala et al., 2018) place points on a hypercube inscribed within a hypersphere, which leads to suboptimal distribution. We also follow a constructive approach, where we use an optimization method for the hyperspherical separation, leading to significantly higher quality embeddings. The eigendecomposition method h-MDS (Sala et al., 2018) takes a graph or tree metric and uses an eigendecomposition of the corresponding distance matrix to generate low-distortion embeddings. However, it collapses nodes within some subtrees to a single point, leading to massive local distortion.

**Deep learning with hyperbolic tree embeddings.** In computer vision, a wide range of works have recently shown the potential and effectiveness of using a hyperbolic embedding space (Khrulkov et al., 2020). Specifically, hierarchical prior knowledge can be embedded in hyperbolic space, after which visual representations can be mapped to the same space and optimized to match this hierarchical organization. (Long et al., 2020) show that such a setup improves hierarchical action recognition, while (Liu et al., 2020) use hierarchies with hyperbolic embeddings for zero-shot learning. Deep visual learning with hyperbolic tree embeddings has furthermore shown to improve image segmentation (Ghadimi Atigh et al., 2022), skin lesion recognition (Yu et al., 2022b), video understanding (Li et al., 2024), hierarchical visual recognition (Ghadimi Atigh et al., 2021; Dhall et al., 2020), hierarchical model interpretation (Gulshad et al., 2023), open set recognition (Dengxiong & Kong, 2023), continual learning (Gao et al., 2023), few-shot learning (Zhang et al., 2022), and more. Since such approaches require freedom in terms of embedding dimensionality, they commonly rely on optimization-based approaches to embed the prior tree-like knowledge. Similar approaches have also been investigated in other domains, from audio (Petermann et al., 2023) and text (Dhingra et al., 2018; Le et al., 2019) to multimodal settings (Hong et al., 2023). In this work, we provide a general-purpose and unconstrained approach for low-distortion embeddings with the option to scale to higher precisions without losing GPU-compatibility.

**Floating point expansions.** Floating point expansions (FPEs) to increase precision in hyperbolic space was proposed by (Yu & De Sa, 2021) and implemented in a PyTorch library (Yu et al., 2022a). However, their methodology is based on older FPE arithmetic definitions and routines by (Priest, 1991; 1992; Richard Shewchuk, 1997). In the field of FPEs, more efficient and stable formulations have been

proposed over the years with improved error guarantees (Joldes et al., 2014; 2015; Muller et al., 2016). In this paper, we build upon the most recent arithmetic framework detailed in (Popescu, 2017). We have implemented this framework for PyTorch and extend its functionality to work with hyperbolic embeddings.

## 3. HS-DTE

**Setting and objective.** We are given a (possibly weighted) tree $T = (V, E)$, where the nodes in $V$ contain the concepts of our hierarchy and the edges in $E$ represent parent-child connections. The goal is to find an embedding $\phi : V \to \mathbb{D}^n$ that accurately captures the semantics of the tree $T$, so where $T$ can be accurately reconstructed from $\phi(V)$. An embedding $\phi$ is evaluated by first defining the graph metric $d_T(u, v)$ on the tree as the length of the shortest path between the nodes $u$ and $v$ and then checking how much $\phi$ distorts this metric. More specifically, for evaluation we use the average relative distortion (Sala et al., 2018), the worst-case distortion (Sarkar, 2011) and the mean average precision (Nickel & Kiela, 2017). Further details on these metrics can be found in Appendix B.

**Constructive solution for hyperbolic embeddings.** The starting point of our method is the Poincaré ball implementation of Sarkar's combinatorial construction (Sarkar, 2011) as outlined by (Sala et al., 2018). A generalized formulation of this approach is outlined in Algorithm 1. The scaling factor $\tau > 0$ is used to scale the tree metric $d_T$. A larger $\tau$ allows for a better use of the curvature of hyperbolic space, theoretically making it easier to find low-distortion embeddings. Lower values can help avoid numerical issues that arise near the boundary of the Poincaré ball. When the dimension of the embedding space satisfies $n \le \log(\Delta(T)) + 1$ and the scaling factor is set to

$$\tau = \frac{1 + \epsilon}{\epsilon} \log \left( 4 \, \Delta(T)^{\frac{1}{n-1}} \right), \qquad (7)$$

with $\Delta(T)$ the maximal degree of $T$, then the construction leads to a worst-case distortion bounded by $1 + \epsilon$, given that the points on the hypersphere are sufficiently uniformly distributed (Sala et al., 2018). When the dimension is $n > \log(\Delta(T)) + 1$, the scaling factor should be $\tau = \Omega(1)$ to achieve this bound on the distortion, so it can no longer be reduced by choosing a higher dimensional embedding space (Sala et al., 2018). The number of bits required for the construction is $\mathcal{O}(\frac{1}{\epsilon} \frac{\ell}{n} \log(\Delta(T)))$ when $n \le \log(\Delta(T)) + 1$ and $\mathcal{O}(\frac{\ell}{\epsilon})$ when $n > \log(\Delta(T)) + 1$, where $\ell$ is the longest path in the tree.

**The difficulty of distributing points on a hypersphere.** The construction in Algorithm 1 provides a nice way of constructing embeddings in $n$-dimensional hyperbolic space

**Algorithm 1** Generalized Sarkar's Delaunay tree embedding

1: **Input:** Tree $T = (V, E)$ and scaling factor $\tau > 0$.
2: **for** $v \in V$ **do**
3: $\quad p \leftarrow \text{parent}(v)$
4: $\quad c_1, \ldots, c_{\deg(v)-1} \leftarrow \text{children}(v)$
5: $\quad$ Reflect $\phi(p)$ with $R_{\phi(v) \to \mathbf{0}}$
6: $\quad$ Generate $\mathbf{x}_1, \ldots, \mathbf{x}_{\deg(v)}$ uniformly distributed points on a hypersphere with radius 1
7: $\quad$ Get rotation matrix $A$ such that $R_{\phi(v) \to \mathbf{0}}\big(\phi(p)\big)$ is aligned with $A\mathbf{x}_{\deg(v)}$ and rotate
8: $\quad$ Scale points by $\gamma = \frac{e^\tau - 1}{e^\tau + 1}$
9: $\quad$ Reflect rotated and scaled points back: $\phi(c_i) \leftarrow R_{\phi(v) \to \mathbf{0}}(\gamma A \mathbf{x}_i), \quad i = 1, \ldots, \deg(v) - 1$
10: **end for**

with arbitrarily low distortion. However, the bound on the distortion for the $\tau$ in Equation 7 is dependent on our ability to generate uniformly distributed points on the $n$-dimensional hypersphere. More specifically, given generated points $\mathbf{x}_1, \ldots, \mathbf{x}_{\Delta(T)}$, the error bound relies on the assumption that

$$\min_{i \neq j} \sin \angle(\mathbf{x}_i, \mathbf{x}_j) \geq \Delta(T)^{-\frac{1}{n-1}}. \quad (8)$$

While this assumption can theoretically always be achieved (Sala et al., 2018), generating points that actually satisfy it is not straightforward. In practice it is important to keep the scaling factor $\tau$ as small as possible, since the required number of bits increases linearly with $\tau$. Increasing the minimal angle beyond the condition in Equation 8 allows for a smaller $\tau$. This problem of maximizing the minimal pairwise angle is commonly known as the Tammes problem (Tammes, 1930; Mettes et al., 2019) and it has been studied extensively, with many analytical and approximate solutions given for various specific combinations of dimensions $n$ and number of points that have to be placed (Cohn, 2024). However, there exists no general closed-form solution that encompasses all possible combinations.

(Sala et al., 2018) propose to generate points by placing them on the vertices of an inscribed hypercube. This approach comes with three limitations. First, the maximum number of points that can be generated with this method is $2^n$, which is limited for small $n$. Second, for most configurations this method results in a sub-optimal distribution, leading to an unnecessarily high requirement on $\tau$. Third, this method depends on finding binary sequences of length $n$ with maximal Hamming distances (see Appendix C), which is in general not an easy problem to solve. Their solution is to use the Hadamard code. This can only be used when the dimension is a power of 2 and at least $\Delta(T)$; a severe restriction, often incompatible with downstream tasks.

**Delaunay tree embeddings with separation.** We propose to improve the construction by distributing the points on the hypersphere in step 6 of Algorithm 1 through optimization. Specifically, we use projected gradient descent to find $\mathbf{x}_1, \ldots, \mathbf{x}_k \in S^{n-1}$ such that

$$\mathbf{x}_1, \ldots, \mathbf{x}_k = \underset{\mathbf{w}_1, \ldots, \mathbf{w}_k \in S^{n-1}}{\arg \min} L(\mathbf{w}_1, \ldots, \mathbf{w}_k), \quad (9)$$

where $L : (S^{n-1})^k \to \mathbb{R}$ is some objective function. Common choices for this objective are the hyperspherical energy functions (Liu et al., 2018), given by

$$E_s(\mathbf{w}_1, \ldots, \mathbf{w}_k) = \begin{cases} \sum_{i=1}^{k} \sum_{j \neq i} \|\mathbf{w}_i - \mathbf{w}_j\|^{-s}, & s > 0, \\ \sum_{i=1}^{k} \sum_{j \neq i} \log\left(\|\mathbf{w}_i - \mathbf{w}_j\|^{-1}\right), & s = 0, \end{cases}$$
$$(10)$$

where $s$ is a nonnegative integer parameterizing this set of functions. Minimizing these objective functions pushes the hyperspherical points apart, leading to a more uniform distribution. However, these objectives are aimed at finding a large mean pairwise angle, allowing for the possibility of having a small minimum pairwise angle. Having a small minimum pairwise angle leads to the corresponding nodes and their descendants being placed too close together, leading to large distortion, as shown in the experiments. Therefore, we advocate the minimal angle maximization (MAM) objective, aimed at maximizing this minimal angle

$$E(\mathbf{w}_1, \ldots, \mathbf{w}_k) = -\sum_{i=1}^{k} \min_{j \neq i} \angle(\mathbf{w}_i, \mathbf{w}_j), \quad (11)$$

which pushes each $\mathbf{w}_i$ away from its nearest neighbour and essentially optimizes directly for the objective of the Tammes problem. We find that this method results in strong separation in terms of the minimal pairwise angle when compared to highly specialized existing methods used for specific cases of the Tammes problem (Cohn, 2024). More importantly, it leads to better separation than the method used in current hyperbolic embeddings (Sala et al., 2018), allowing for the use of a smaller $\tau$. Moreover, this optimization method places no requirements on the dimension, making it a suitable choice for downstream tasks. We refer to the resulting construction as the highly separated Delaunay tree embedding (HS-DTE).

When performing the construction using MAM, the output of the optimization can be cached and reused each time a node with the same degree is encountered. Using this approach, the worst-case number of times that a solution to equation 9 has to be found is $\mathcal{O}(\sqrt{N})$ as shown by Theorem 3.1.

**Theorem 3.1.** *The worst-case number of optimizations $p$ that has to be performed when embedding a tree with the*

*combinatorial construction in Algorithm 1 with any objective using caching is*

$$p \leq \left\lceil \frac{1}{2}(1 + \sqrt{16N - 15}) \right\rceil. \qquad (12)$$

*Proof.* See Appendix D. □

In practice we find the number of optimizations to be lower due to frequent occurrence of low degree nodes for which cached points can be used, as shown in Appendix L.

**MAM optimization details.** MAM is an easily optimizable objective, that we train using projected gradient descent for 450 iterations with a learning rate of 0.01, reduced by a factor of 10 every 150 steps, for every configuration. This optimization can generally be performed in mere seconds which, if necessary, could be further optimized through hyperparameter configurations, early stopping, parallelization or hardware acceleration. As a result, the increase in computation time of our method compared to (Sala et al., 2018) is minimal. Moreover, when compared to methods such as Poincaré embeddings (Nickel & Kiela, 2017) which use stochastic gradient descent to directly optimize the embeddings, we find that our method is orders of magnitude faster, while avoiding the need for costly hyperparameter tuning.

## 4. HypFPE: High-precision GPU-compatible Hyperbolic Embeddings

While hyperbolic space enjoys numerous potential benefits, it is prone to numerical error when using floating point arithmetic. Especially as points move away from the origin, floating point arithmetic struggles to accurately represent or perform computations with these points. For larger values of $\tau$ or maximal path lengths $\ell$, the embeddings generated by the construction often end up in this problematic region of the Poincaré ball. As such, the precision required for hyperbolic embeddings is often larger than the precision provided by the floating point formats supported on GPUs. Increased precision can be attained by switching to arbitrary precision arithmetic. However, this makes the result incompatible with existing deep learning libraries.

Here, we propose HypFPE, a method to increase the precision of constructive hyperbolic approaches through floating point expansion (FPE) arithmetic. In this framework, numbers are represented as unevaluated sums of floating point numbers, typically of a fixed number of bits $b$. In other words, a number $f \in \mathbb{R}$ is represented by a floating point expansion $\tilde{f}$ as

$$f \approx \tilde{f} = \sum_{i=1}^{t} \tilde{f}_i, \qquad (13)$$

where the $\tilde{f}_i$ are floating point numbers with a fixed number of bits and where $t$ is the number of terms that the floating point expansion $\tilde{f}$ consists of. Each term $\tilde{f}_i$ is in the form of a GPU supported float format, such as float16, float32 or float64. Moreover, to ensure that this representation is unique and uses bits efficiently, it is constrained to be ulp-nonoverlapping (Popescu, 2017).

**Definition 4.1.** A floating point expansion $\tilde{f} = \tilde{f}_1 + \ldots + \tilde{f}_t$ is ulp-nonoverlapping if for all $2 \leq i \leq t$, $|\tilde{f}_i| \leq \text{ulp}(\tilde{f}_{i-1})$, where $\text{ulp}(\tilde{f}_{i-1})$ is the *unit in the last place* of $\tilde{f}_{i-1}$.

A ulp-nonoverlapping FPE consisting of $t$ terms each with $b$ bits precision has at worst $t(b-1) + 1$ bits of precision, since exactly $t - 1$ overlapping bits can occur. The corresponding arithmetic requires completely different routines for computing basic operations, many of which have been introduced by (Joldes et al., 2014; 2015; Muller et al., 2016; Popescu, 2017). An overview of these routines can be found in Appendix N. For an overview of the error guarantees we refer to (Popescu, 2017). Each of these routines can be defined using ordinary floating point operations that exist for tensors in standard tensor libraries such as PyTorch, which are completely GPU compatible. Here, we have generalized all routines to tensor operations and implemented them in PyTorch by adding an extra dimension to each tensor containing the terms of the floating point expansion.

**Applying FPEs to the construction.** In the constructive method the added precision is warranted whenever numerical errors lead to large deviations with respect to the hyperbolic metric. In other words, if we have some $\mathbf{x} \in \mathbb{D}^n$ and its floating point representation $\tilde{\mathbf{x}}$, then it makes sense to increase the precision if

$$d_{\mathbb{D}}(\mathbf{x}, \tilde{\mathbf{x}}) \gg 0. \qquad (14)$$

For the Poincaré ball, this is the case whenever $\mathbf{x}$ lies somewhere close to the boundary of the ball. In our construction, this means that generation and rotation of points on the unit hypersphere can be performed in normal floating point arithmetic, since the representation error in terms of $d_{\mathbb{D}}$ will be negligible. However, for large $\tau$, the scaling of the hypersphere points and the hyperspherical inversion require increased precision as these map points close to the boundary of $\mathbb{D}^n$. Specifically, steps 5, 8 and 9 of Algorithm 1 may require increased precision. Note that these operations can be performed using the basic operation routines shown in Appendix N. From the basic operations, more complicated nonlinear operations can be defined through the Taylor series approximations that are typically used for floating point arithmetic. To compute the distortion of the resulting embeddings, the distances between the embedded nodes must be computed either through the inverse hyperbolic cosine formulation of Equation 2 or through the inverse hyperbolic tangent formulation of Equation 3. We show how to accurately compute distances using either formulation.

## 4.1. The Inverse Hyperbolic Cosine Formulation

For Equation 2, normal floating point arithmetic may cause the denominator inside the argument of $\cosh^{-1}$ to become 0 due to rounding. To solve this, we can use FPE arithmetic to compute the argument of $\cosh^{-1}$ and then approximate the distance by applying $\cosh^{-1}$ to the largest magnitude term of the FPE. This allows accurate computation of distances even for points near the boundary of the Poincaré ball, as shown by Theorem 4.2 and Proposition 4.3.

**Theorem 4.2.** *Given* $\mathbf{x}, \mathbf{y} \in \mathbb{D}^n$ *with* $||\mathbf{x}|| < 1 - \epsilon^{t-1}$ *and* $||\mathbf{y}|| < 1 - \epsilon^{t-1}$, *an approximation* $d$ *to equation 2 can be computed with FPE representations with* $t$ *terms and with a largest magnitude approximation to* $\cosh^{-1}$ *such that, for some small* $\epsilon^* > 0$,

$$\left| d - \cosh^{-1}\left( 1 + 2\frac{||\mathbf{x} - \mathbf{y}||^2}{(1 - ||\mathbf{x}||^2)(1 - ||\mathbf{y}||^2)} \right) \right| < \epsilon^*. \quad (15)$$

*Proof.* See Appendix E. $\qquad\square$

**Proposition 4.3.** *The range of the inverse hyperbolic tangent formulation increases linearly in the number of terms* $t$ *of the FPEs being used.*

*Proof.* See Appendix F. $\qquad\square$

Theorem 4.2 shows that we can accurately compute distances on a larger domain than with normal floating point arithmetic. Proposition 4.3 shows that the effective radius of the Poincaré ball in which we can represent points and compute distances increases linearly in the number of terms of our FPE expansions. Therefore, this effective radius increases linearly with the number of bits. The same holds for arbitrary precision floating point arithmetic, so FPE expansions require a similar number of bits for constructive methods as arbitrary precision floating point arithmetic.

## 4.2. The Inverse Hyperbolic Tangent Formulation

For Equation 3, the difficulty lies in the computation of $\tanh^{-1}$. With normal floating point arithmetic, due to rounding errors, this function can only be evaluated on $[-1 + \epsilon, 1 - \epsilon]$, where $\epsilon$ is the machine precision. This severely limits the range of values, i.e., distances, that we can compute. Therefore, we need to be able to compute the inverse hyperbolic tangent with FPEs. Inspired by (Felker & musl Contributors, 2024), we propose a new routine for this computation, given in Algorithm 13 of Appendix N. Here, we approximate the logarithm in steps 7 and 9 as $\log(\tilde{f}) \approx \log(\tilde{f}_1)$, which is accurate enough for our purposes. This algorithm can be used to accurately approximate $\tanh^{-1}$ while extending the range linearly in the number of terms $t$ as shown by Theorem 4.4 and Proposition 4.5.

**Theorem 4.4.** *Given a ulp-nonoverlapping FPE* $x = \sum_{i=1}^t x_i \in [-1 + \epsilon^{t-1}, 1 - \epsilon^{t-1}]$ *consisting of floating point numbers with a precision* $b > t$, *Algorithm 13 leads to an approximation* $y$ *of the inverse hyperbolic tangent of* $x$ *that, for small* $\epsilon^* > 0$, *satisfies*

$$|y - \tanh^{-1}(x)| \leq \epsilon^*. \quad (16)$$

*Proof.* See Appendix G. $\qquad\square$

**Proposition 4.5.** *The range of algorithm 13 increases linearly in the number of terms* $t$.

*Proof.* See Appendix H. $\qquad\square$

Based on these results, either formulation could be a good choice for computing distances with FPEs. In practice, we find that the $\tanh^{-1}$ formulation leads to larger numerical errors, which is likely due to catastrophic cancellation errors in the dot product that is performed in Equation 4. Therefore, we use the $\cosh^{-1}$ formulation in our experiments.

## 5. Experiments

### 5.1. Ablations

**Minimal angle maximization.** To test how well the proposed methods for hyperspherical separation perform, we generate points $\mathbf{w}_1, \ldots, \mathbf{w}_k$ on an 8-dimensional hypersphere for various numbers of points $k$ and compute the minimal pairwise angle $\min_{i \neq j} \angle(\mathbf{w}_i, \mathbf{w}_j)$. We compare to the Hadamard generation method from (Sala et al., 2018) and the method that is used in their implementation, which precomputes 1000 points using the method from (Lovisolo & Da Silva, 2001) and samples from these precomputed points. Note that a power of 2 is chosen for the dimension to be able to make a fair comparison to the Hadamard construction, since this method cannot be used otherwise. The results are shown in Figure 1a. These results show that our MAM indeed leads to high separation in terms of the minimal pairwise angle, that the precomputed approach leads to poor separation and that the Hadamard method only performs moderately well when the number of points required is close to the dimension of the space.

To verify that this minimal pairwise angle is important for the quality of the construction, we perform the construction on a binary tree with a depth of 8 edges using each of the hypersphere generation methods. The construction is performed in 10 dimensions except for the Hadamard method, since this cannot generate 10 dimensional points. Additional results for dimensions 4, 7 and 20 are shown in Appendix I. Each method is applied using float32 representations and a scaling factor of $\tau = 1.33$. The results are shown in Table 1. These findings support our hypothesis

*Table 1.* **Comparing hyperspherical separation** methods for the constructive hyperbolic embedding of a binary tree with a depth of 8 edges when using float32 representations (24 bits precision) in 10 dimensions. Note that the Hadamard method (‡) cannot be applied in 10 dimensions, so there 8 is used instead.

| Method | dim | $D_{ave}$ | $D_{wc}$ | MAP |
|---|---|---|---|---|
| (Sala et al., 2018) ‡ | 8 | 0.734 | 1143 | 0.154 |
| (Sala et al., 2018) ⋆ | 10 | 0.361 | 18.42 | 0.998 |
| $E_0$ | 10 | 0.219 | 1.670 | **1.000** |
| $E_1$ | 10 | 0.204 | 1.686 | **1.000** |
| $E_2$ | 10 | 0.190 | 1.642 | **1.000** |
| MAM | 10 | **0.188** | **1.635** | **1.000** |

that the minimal pairwise angle is important for generating high quality embeddings and that the MAM is an excellent objective function for performing the separation.

**FPEs versus standard floating points.** To demonstrate the importance of using FPEs for increasing precision, we perform the construction on a phylogenetic tree expressing the genetic heritage of mosses in urban environments (Hofbauer et al., 2016), made available by (Sanderson et al., 1994), using various precisions. This tree has a maximum path length $\ell = 30$, which imposes sharp restrictions on the value of $\tau$ that we can choose before encountering numerical errors. We perform the construction either with normal floating point arithmetic using the usual GPU-supported float formats or with FPEs, using multiple float64 terms. The scaling factor $\tau$ is chosen to be close to the threshold where numerical problems appear in order to obtain optimal results for the given precision. The results in terms of $D_{wc}$ are shown in Figure 1b. As can be seen from these results, around 100 bits of precision are needed to obtain decent results, which can be achieved using FPEs with 2 float64 terms. Without FPE expansions, the largest GPU-compatible precision is 53 bits, obtained by using float64. This precision yields a $D_{wc}$ of 9.42, which is quite poor. These results illustrate the importance of FPEs for high quality GPU-compatible embeddings.

### 5.2. Embedding Complete $m$-ary Trees

To demonstrate the strong performance of the combinatorial constructions compared to other methods, we perform embeddings on several complete $m$-ary trees with a max path length of $\ell = 8$ and branching factors $m = 3, 5, 7$. Due to the small $\ell$, each experiment can be performed with normal floating point arithmetic using float64 representations. We compare our method with Poincaré embeddings (PE) (Nickel & Kiela, 2017), hyperbolic entailment cones (HEC) (Ganea et al., 2018), distortion optimization (DO) (Sala et al., 2018; Yu et al., 2022b), h-MDS (Sala et al.,

2018) and the combinatorial method with Hadamard (Sala et al., 2018) or precomputed hyperspherical points (Lovisolo & Da Silva, 2001). For the constructive methods and for h-MDS, a larger scaling factor improves performance, so we use $\tau = 5$. For DO we find that increasing the scaling factor does not improve performance, so we use $\tau = 1.0$. PE and HEC are independent of the scaling factor.

The results on the various trees in 10 dimensions are shown in table 2 and additional results for dimensions 4, 7 and 20 are shown in Appendix J. These illustrate the strength of the combinatorial constructions. The optimization methods PE, HEC and DO perform relatively poor for all evaluation metrics. This performance could be increased through hyperparameter tuning and longer training. However, the results will not come close to those of the other methods. The h-MDS method performs well in terms of $D_{ave}$, but very poorly on $D_{wc}$ and MAP. This is because h-MDS collapses leaf nodes, leading to massive local distortion within the affected subtrees. However, between subtrees this distortion is much smaller, explaining the low $D_{ave}$. Figure 2 illustrates the issue with h-MDS and the superiority of our approach. Each of the white squares in the h-MDS plot corresponds to a collapsed subtree, which renders the embeddings unusable for downstream tasks since nearby leaf nodes cannot be distinguished. We conclude that HS-DTE obtains the strongest embeddings overall.

### 5.3. Embedding Phylogenetic Trees

Lastly, we compare hyperbolic embeddings on phylogenetic trees. Moreover, we show how adding HypFPE to our method and the other combinatorial methods increases the embedding quality when requiring GPU-compatibility. The phylogenetic trees describe mosses (Hofbauer et al., 2016), weevils (Marvaldi et al., 2002), the European carnivora (Roquet et al., 2014), and lichen (Zhao et al., 2016), obtained from (McTavish et al., 2015). The latter two trees are weighted trees which can be embedded by adjusting the scaling in step 8 of Algorithm 1. Each of the embeddings is performed in 10-dimensional space. Other dimensions are given in Appendix K. The h-MDS method and the combinatorial constructions are performed with the largest $\tau$ that can be used with the given precision. The results are shown in Table 3. When using float64, HS-DTE outperforms each of the optimization-based methods and the other combinatorial approaches from (Sala et al., 2018). While h-MDS obtains high average distortion, it collapses entire subtrees, leading to massive local distortion. Therefore, the HS-DTE embeddings are of the highest quality. When adding HypFPE on top of the combinatorial approaches, all performances go up, with the combination of HS-DTE and HypFPE leading to the best performance on both $D_{ave}$ and $D_{wc}$. Additional results on graph-like data are shown in Appendix M.

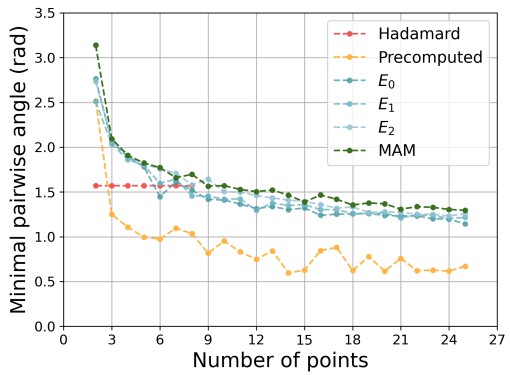

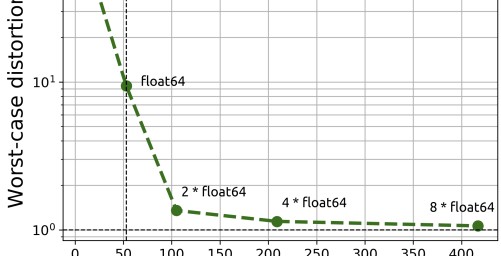

(a) Minimal hyperspherical energy ablation.

(b) Floating point expansion ablation.

*Figure 1.* **Ablation studies on our construction and floating point expansion.** (a) Minimal pairwise angle ($\uparrow$) of the hyperspherical points generated in step 6 of Algorithm 1 using the various generation methods. The dimension of the space is set to 8, so the Hadamard method cannot generate more than 8 points. The MAM objective consistently leads to a higher separation angle. (b) The worst-case distortion ($\downarrow$, $D_{wc}$) of the constructed embedding of the phylogenetic tree with the maximal admissable $\tau$ given the number of bits. The vertical dashed line shows the limit with standard GPU floating point formats (float64). The horizontal dashed line is the best possible result $D_{wc} = 1$. FPE representations are required to get high quality embeddings without losing GPU-compatibility.

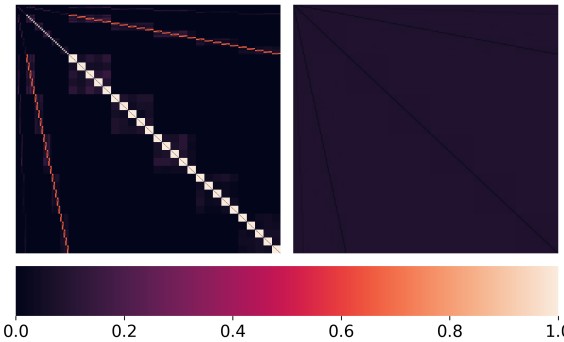

*Figure 2.* **Pairwise relative distortions of h-MDS (left) and HS-DTE (right)** applied to the 5-ary tree with a scaling factor $\tau = 5.0$. Axes are ordered using a breadth-first search of the tree. Embeddings obtained with h-MDS suffer from leaf node collapse.

## 6. Conclusion

In this paper we introduce HS-DTE, a novel way of constructively embedding trees in hyperbolic space, which uses an optimization approach to effectively separate points on a hypersphere. Empirically, we show that HS-DTE outperforms existing methods, while maintaining the computational efficiency of the combinatorial approaches. We also introduce HypFPE, a framework for floating point expansion arithmetic on tensors, which is adapted to extend the effective radius of the Poincaré ball. This framework can be used to increase the precision of computations, while benefiting from hardware acceleration, paving the way for highly accurate hyperbolic neural networks. It can be added on top of any of the combinatorial methods, leading to low-distortion *and* GPU-compatible hyperbolic tree embeddings.

## Acknowledgements

Max van Spengler gratefully acknowledges the University of Amsterdam Data Science Centre for financial support.

## Impact Statement

This paper presents work whose goal is to advance the field of Machine Learning. There are many potential societal consequences of our work, none which we feel must be specifically highlighted here.

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

*Table 2.* **Comparison of hyperbolic embedding algorithms on $m$-ary trees** with a maximum path length of $\ell = 8$. The h-MDS method is represented by †. The ‡ method is the combinatorial construction with the hyperspherical points being generated using the Hadamard construction, whereas the ⋆ method samples hyperspherical points from the precomputed points generated with the hyperspherical separation method from (Lovisolo & Da Silva, 2001). The h-MDS method outperforms the other methods in terms of $D_{ave}$, but collapses nodes, leading to NaN values of the $D_{wc}$ and making the embeddings unusable. HS-DTE has the second best $D_{ave}$ and outperforms all methods in terms of $D_{wc}$. Each combinatorial construction has a perfect MAP.

| | 3-tree | | | 5-tree | | | 7-tree | | |
| --- | --- | --- | --- | --- | --- | --- | --- | --- | --- |
| | $D_{ave}$ | $D_{wc}$ | MAP | $D_{ave}$ | $D_{wc}$ | MAP | $D_{ave}$ | $D_{wc}$ | MAP |
| (Nickel & Kiela, 2017) | 0.17 | 169 | 0.8 | 0.31 | NaN | 0.58 | 0.84 | NaN | 0.24 |
| (Ganea et al., 2018) | 0.51 | 184 | 0.27 | 0.81 | 604 | 0.24 | 0.96 | 788 | 0.15 |
| (Yu et al., 2022b) | 0.16 | 31.9 | 0.57 | 0.52 | 545 | 0.30 | 0.93 | 3230 | 0.05 |
| (Sala et al., 2018) † | **0.03** | NaN | 0.52 | **0.04** | NaN | 0.1 | **0.03** | NaN | 0.05 |
| (Sala et al., 2018) ‡ | 0.11 | 1.14 | **1.00** | 0.12 | 1.14 | **1.00** | 0.12 | 1.14 | **1.00** |
| (Sala et al., 2018) ⋆ | 0.09 | 1.18 | **1.00** | 0.13 | 1.30 | **1.00** | 0.13 | 1.31 | **1.00** |
| HS-DTE | 0.06 | **1.07** | **1.00** | 0.09 | **1.09** | **1.00** | 0.10 | **1.12** | **1.00** |

*Table 3.* **Comparison of hyperbolic embedding algorithms on various trees.** † represents h-MDS, ‡ the construction with Hadamard hyperspherical points and ⋆ the construction with points sampled from a set precomputed with (Lovisolo & Da Silva, 2001). The best float64 performance is underlined and the best FPE performance is in bold. All embeddings are performed in a 10-dimensional space. Hadamard generation cannot be used for the mosses tree, since it has a $\Delta(T)$ greater than 8. Distortion optimization (Yu et al., 2022b) does not converge for the lichen tree due to large variation in edge weights. Overall, combining HypFPE and HS-DTE works best.

| | Precision bits | Mosses | | Weevils | | Carnivora | | Lichen | |
| --- | --- | --- | --- | --- | --- | --- | --- | --- | --- |
| | | $D_{ave}$ | $D_{wc}$ | $D_{ave}$ | $D_{wc}$ | $D_{ave}$ | $D_{wc}$ | $D_{ave}$ | $D_{wc}$ |
| (Nickel & Kiela, 2017) | 53 | 0.68 | 44350 | 0.45 | NaN | 0.96 | NaN | 151 | NaN |
| (Ganea et al., 2018) | 53 | 0.90 | 1687 | 0.77 | 566 | 0.99 | NaN | 162 | NaN |
| (Yu et al., 2022b) | 53 | 0.83 | 163 | 0.57 | 79.8 | 0.99 | NaN | - | - |
| (Sala et al., 2018) † | 53 | 0.04 | NaN | 0.06 | NaN | 0.11 | NaN | 0.13 | NaN |
| (Sala et al., 2018) ‡ | 53 | - | - | 0.79 | 330 | 0.26 | 35.2 | 0.49 | 79.6 |
| (Sala et al., 2018) ⋆ | 53 | 0.78 | 122 | 0.54 | 34.3 | 0.23 | 18.8 | 0.55 | 101 |
| HS-DTE | 53 | 0.40 | 9.42 | 0.27 | 2.03 | 0.12 | 11.7 | 0.30 | 23.5 |
| HypFPE + (Sala et al., 2018) ‡ | 417 | - | - | 0.07 | 1.09 | 0.05 | 6.76 | 0.12 | 43.4 |
| HypFPE + (Sala et al., 2018) ⋆ | 417 | 0.08 | 1.14 | 0.05 | 1.11 | **0.03** | 4.87 | 0.11 | 6.42 |
| HypFPE + HS-DTE | 417 | **0.04** | **1.06** | **0.03** | **1.04** | **0.03** | **2.03** | **0.05** | **3.30** |

De Nooy, W., Mrvar, A., and Batagelj, V. *Exploratory social network analysis with Pajek: Revised and expanded edition for updated software*, volume 46. Cambridge university press, 2018.

Dengxiong, X. and Kong, Y. Ancestor search: Generalized open set recognition via hyperbolic side information learning. In *Proceedings of the IEEE/CVF Winter Conference on Applications of Computer Vision*, pp. 4003–4012, 2023.

Desai, K., Nickel, M., Rajpurohit, T., Johnson, J., and Vedantam, S. R. Hyperbolic image-text representations. In *International Conference on Machine Learning*, pp. 7694–7731. PMLR, 2023.

Dhall, A., Makarova, A., Ganea, O., Pavllo, D., Greeff, M., and Krause, A. Hierarchical image classification using entailment cone embeddings. In *Proceedings of the IEEE/CVF conference on computer vision and pattern recognition workshops*, pp. 836–837, 2020.

Dhingra, B., Shallue, C., Norouzi, M., Dai, A., and Dahl, G. Embedding text in hyperbolic spaces. In *Proceedings of the Twelfth Workshop on Graph-Based Methods for Natural Language Processing*, pp. 59–69, 2018.

Felker, R. and musl Contributors. musl libc: A lightweight implementation of the standard library for linux systems, 2024. URL https://musl.libc.org. Version 1.2.5, retrieved on September 30, 2024.

Freeman, L. The development of social network analysis. *A Study in the Sociology of Science*, 1(687):159–167, 2004.

Ganea, O., Bécigneul, G., and Hofmann, T. Hyperbolic entailment cones for learning hierarchical embeddings. In *International conference on machine learning*, pp. 1646–1655. PMLR, 2018.

Gao, Z., Xu, C., Li, F., Jia, Y., Harandi, M., and Wu, Y. Exploring data geometry for continual learning. In *Proceedings of the IEEE/CVF conference on computer vision and pattern recognition*, pp. 24325–24334, 2023.

Ghadimi Atigh, M., Keller-Ressel, M., and Mettes, P. Hyperbolic busemann learning with ideal prototypes. *Advances in neural information processing systems*, 34:103–115, 2021.

Ghadimi Atigh, M., Schoep, J., Acar, E., Van Noord, N., and Mettes, P. Hyperbolic image segmentation. In *Proceedings of the IEEE/CVF conference on computer vision and pattern recognition*, pp. 4453–4462, 2022.

Goh, K.-I., Cusick, M. E., Valle, D., Childs, B., Vidal, M., and Barabási, A.-L. The human disease network. *Proceedings of the National Academy of Sciences*, 104 (21):8685–8690, 2007.

Gulshad, S., Long, T., and van Noord, N. Hierarchical explanations for video action recognition. In *Proceedings of the IEEE/CVF Conference on Computer Vision and Pattern Recognition*, pp. 3703–3708, 2023.

Hofbauer, W. K., Forrest, L. L., Hollingsworth, P. M., and Hart, M. L. Preliminary insights from dna barcoding into the diversity of mosses colonising modern building surfaces. *Bryophyte Diversity and Evolution*, 38(1):1–22, 2016.

Hong, J., Hayder, Z., Han, J., Fang, P., Harandi, M., and Petersson, L. Hyperbolic audio-visual zero-shot learning. In *Proceedings of the IEEE/CVF international conference on computer vision*, pp. 7873–7883, 2023.

Joldes, M., Muller, J.-M., and Popescu, V. On the computation of the reciprocal of floating point expansions using an adapted Newton-Raphson iteration. In *2014 IEEE 25th International Conference on Application-Specific Systems, Architectures and Processors*, pp. 63–67. IEEE, 2014.

Joldes, M., Marty, O., Muller, J.-M., and Popescu, V. Arithmetic algorithms for extended precision using floating-point expansions. *IEEE Transactions on Computers*, 65 (4):1197–1210, 2015.

Kapli, P., Yang, Z., and Telford, M. J. Phylogenetic tree building in the genomic age. *Nature Reviews Genetics*, 21(7):428–444, 2020.

Khrulkov, V., Mirvakhabova, L., Ustinova, E., Oseledets, I., and Lempitsky, V. Hyperbolic image embeddings. In *Proceedings of the IEEE/CVF conference on computer vision and pattern recognition*, pp. 6418–6428, 2020.

Kolyvakis, P., Kalousis, A., and Kiritsis, D. Hyperbolic knowledge graph embeddings for knowledge base completion. In *The Semantic Web: 17th International Conference, ESWC 2020, Heraklion, Crete, Greece, May 31–June 4, 2020, Proceedings 17*, pp. 199–214. Springer, 2020.

Le, M., Roller, S., Papaxanthos, L., Kiela, D., and Nickel, M. Inferring concept hierarchies from text corpora via hyperbolic embeddings. *arXiv preprint arXiv:1902.00913*, 2019.

Li, Y.-L., Wu, X., Liu, X., Wang, Z., Dou, Y., Ji, Y., Zhang, J., Li, Y., Lu, X., Tan, J., et al. From isolated islands to pangea: Unifying semantic space for human action understanding. In *Proceedings of the IEEE/CVF Conference on Computer Vision and Pattern Recognition*, pp. 16582–16592, 2024.

Liu, S., Chen, J., Pan, L., Ngo, C.-W., Chua, T.-S., and Jiang, Y.-G. Hyperbolic visual embedding learning for zero-shot recognition. In *Proceedings of the IEEE/CVF conference on computer vision and pattern recognition*, pp. 9273–9281, 2020.

Liu, W., Lin, R., Liu, Z., Liu, L., Yu, Z., Dai, B., and Song, L. Learning towards minimum hyperspherical energy. *Advances in neural information processing systems*, 31, 2018.

Long, T., Mettes, P., Shen, H. T., and Snoek, C. G. Searching for actions on the hyperbole. In *Proceedings of the IEEE/CVF Conference on Computer Vision and Pattern Recognition*, pp. 1141–1150, 2020.

Lovisolo, L. and Da Silva, E. Uniform distribution of points on a hyper-sphere with applications to vector bit-plane encoding. *IEE Proceedings-Vision, Image and Signal Processing*, 148(3):187–193, 2001.

MacWilliams and Sloane. The theory of error-correcting codes. *Elsevier Science Publishers BV google schola*, 2: 39–47, 1977.

Marvaldi, A. E., Sequeira, A. S., O'Brien, C. W., and Farrell, B. D. Molecular and morphological phylogenetics of weevils (coleoptera, curculionoidea): do niche shifts accompany diversification? *Systematic biology*, 51(5): 761–785, 2002.

McTavish, E. J., Hinchliff, C. E., Allman, J. F., Brown, J. W., Cranston, K. A., Holder, M. T., Rees, J. A., and Smith,

S. A. Phylesystem: a git-based data store for community-curated phylogenetic estimates. *Bioinformatics*, 31(17): 2794–2800, 2015.

Mettes, P., Van der Pol, E., and Snoek, C. Hyperspherical prototype networks. *Advances in neural information processing systems*, 32, 2019.

Mettes, P., Ghadimi Atigh, M., Keller-Ressel, M., Gu, J., and Yeung, S. Hyperbolic deep learning in computer vision: A survey. *International Journal of Computer Vision*, pp. 1–25, 2024.

Miller, G. A. Wordnet: a lexical database for english. *Communications of the ACM*, 38(11):39–41, 1995.

Muller, J.-M., Popescu, V., and Tang, P. T. P. A new multiplication algorithm for extended precision using floating-point expansions. In *2016 IEEE 23nd Symposium on Computer Arithmetic (ARITH)*, pp. 39–46. IEEE, 2016.

Nickel, M. and Kiela, D. Poincaré embeddings for learning hierarchical representations. *Advances in neural information processing systems*, 30, 2017.

Nickel, M. and Kiela, D. Learning continuous hierarchies in the Lorentz model of hyperbolic geometry. In *International conference on machine learning*, pp. 3779–3788. PMLR, 2018.

Padial, J. M., Miralles, A., De la Riva, I., and Vences, M. The integrative future of taxonomy. *Frontiers in zoology*, 7:1–14, 2010.

Pal, A., van Spengler, M., di Melendugno, G. M. D., Flaborea, A., Galasso, F., and Mettes, P. Compositional entailment learning for hyperbolic vision-language models. *International Conference on Learning Representations*, 2025.

Peng, W., Varanka, T., Mostafa, A., Shi, H., and Zhao, G. Hyperbolic deep neural networks: A survey. *IEEE Transactions on pattern analysis and machine intelligence*, 44 (12):10023–10044, 2021.

Petermann, D., Wichern, G., Subramanian, A., and Le Roux, J. Hyperbolic audio source separation. In *ICASSP 2023-2023 IEEE International Conference on Acoustics, Speech and Signal Processing (ICASSP)*, pp. 1–5. IEEE, 2023.

Popescu, V. *Towards fast and certified multiple-precision librairies*. PhD thesis, Université de Lyon, 2017.

Priest, D. M. Algorithms for arbitrary precision floating point arithmetic. *University of California, Berkeley*, 1991.

Priest, D. M. *On properties of floating point arithmetics: numerical stability and the cost of accurate computations*. University of California, Berkeley, 1992.

Richard Shewchuk, J. Adaptive precision floating-point arithmetic and fast robust geometric predicates. *Discrete & Computational Geometry*, 18:305–363, 1997.

Roquet, C., Lavergne, S., and Thuiller, W. One tree to link them all: a phylogenetic dataset for the european tetrapoda. *PLoS currents*, 6, 2014.

Saff, E. B. and Kuijlaars, A. B. Distributing many points on a sphere. *The mathematical intelligencer*, 19:5–11, 1997.

Sala, F., De Sa, C., Gu, A., and Ré, C. Representation trade-offs for hyperbolic embeddings. In *International Conference on Machine Learning*, pp. 4460–4469. PMLR, 2018.

Sanderson, M. J., Donoghue, M. J., Piel, W., and Eriksson, T. Treebase: a prototype database of phylogenetic analyses and an interactive tool for browsing the phylogeny of life. *American Journal of Botany*, 81(6):183, 1994.

Sarkar, R. Low distortion delaunay embedding of trees in hyperbolic plane. In *International Symposium on Graph Drawing*, pp. 355–366. Springer, 2011.

Sonthalia, R. and Gilbert, A. Tree! i am no tree! i am a low dimensional hyperbolic embedding. *Advances in Neural Information Processing Systems*, 33:845–856, 2020.

Tammes, P. M. L. On the origin of number and arrangement of the places of exit on the surface of pollen-grains. *Recueil des travaux botaniques néerlandais*, 27(1):1–84, 1930.

Tifrea, A., Bécigneul, G., and Ganea, O.-E. Poincaré glove: Hyperbolic word embeddings. *arXiv preprint arXiv:1810.06546*, 2018.

Van Spengler, M., Berkhout, E., and Mettes, P. Poincaré resnet. In *Proceedings of the IEEE/CVF International Conference on Computer Vision*, pp. 5419–5428, 2023.

Yang, Z., Yang, D., Dyer, C., He, X., Smola, A., and Hovy, E. Hierarchical attention networks for document classification. In *Proceedings of the 2016 conference of the North American chapter of the association for computational linguistics: human language technologies*, pp. 1480–1489, 2016.

Yu, T. and De Sa, C. M. Representing hyperbolic space accurately using multi-component floats. *Advances in Neural Information Processing Systems*, 34:15570–15581, 2021.

Yu, T., Guo, W., Li, J. C., Yuan, T., and De Sa, C. Mctensor: A high-precision deep learning library with multi-component floating-point. *arXiv preprint arXiv:2207.08867*, 2022a.

Yu, Z., Nguyen, T., Gal, Y., Ju, L., Chandra, S. S., Zhang, L., Bonnington, P., Mar, V., Wang, Z., and Ge, Z. Skin lesion recognition with class-hierarchy regularized hyperbolic embeddings. In *International conference on medical image computing and computer-assisted intervention*, pp. 594–603. Springer, 2022b.

Zhang, B., Jiang, H., Feng, S., Li, X., Ye, Y., and Ye, R. Hyperbolic knowledge transfer with class hierarchy for few-shot learning. In *IJCAI*, pp. 3723–3729, 2022.

Zhao, X., Leavitt, S. D., Zhao, Z. T., Zhang, L. L., Arup, U., Grube, M., Pérez-Ortega, S., Printzen, C., Śliwa, L., Kraichak, E., et al. Towards a revised generic classification of lecanoroid lichens (lecanoraceae, ascomycota) based on molecular, morphological and chemical evidence. *Fungal Diversity*, 78:293–304, 2016.

## A. Geodesic hyperplane reflections

In this paper we will make use of reflections in geodesic hyperplanes through the origin to align points on a hypersphere centered at the origin with some existing point on the hypersphere. More specifically, if we have points $\mathbf{w}, \mathbf{z} \in \mathbb{D}^n$ with $||\mathbf{w}|| = ||\mathbf{z}||$ and we want to reflect $\mathbf{w}$ to $\mathbf{z}$, then we can use a Householder reflection with $\mathbf{v} = \frac{(\mathbf{z}-\mathbf{w})}{||\mathbf{z}-\mathbf{w}||}$, so

$$R_{\mathbf{w}\to\mathbf{z}}(\mathbf{y}) = \left( I_n - \frac{2(\mathbf{z}-\mathbf{w})(\mathbf{z}-\mathbf{w})^T}{||\mathbf{z}-\mathbf{w}||^2} \right)\mathbf{y}. \tag{17}$$

To see that this maps $\mathbf{w}$ to $\mathbf{z}$, we can simply enter $\mathbf{w}$ into this map to see that

$$R_{\mathbf{w}\to\mathbf{z}}(\mathbf{w}) = \left( I_n - \frac{2(\mathbf{z}-\mathbf{w})(\mathbf{z}-\mathbf{w})^T}{||\mathbf{z}-\mathbf{w}||^2} \right)\mathbf{w} \tag{18}$$

$$= \mathbf{w} - \frac{2(\langle \mathbf{z}, \mathbf{w}\rangle - ||\mathbf{w}||^2)}{||\mathbf{z}||^2 - 2\langle \mathbf{z}, \mathbf{w}\rangle + ||\mathbf{w}||^2}(\mathbf{z}-\mathbf{w}) \tag{19}$$

$$= \mathbf{w} + \frac{||\mathbf{z}||^2 - 2\langle \mathbf{z}, \mathbf{w}\rangle + ||\mathbf{w}||^2 + ||\mathbf{w}||^2 - ||\mathbf{z}||^2}{||\mathbf{z}||^2 - 2\langle \mathbf{z}, \mathbf{w}\rangle + ||\mathbf{w}||^2}(\mathbf{z}-\mathbf{w}) \tag{20}$$

$$= \mathbf{w} + \left( 1 + \frac{||\mathbf{w}||^2 - ||\mathbf{z}||^2}{||\mathbf{z}||^2 - 2\langle \mathbf{z}, \mathbf{w}\rangle + ||\mathbf{w}||^2} \right)(\mathbf{z}-\mathbf{w}) \tag{21}$$

$$= \mathbf{w} + \mathbf{z} - \mathbf{w} = \mathbf{z}. \tag{22}$$

We make use of reflections in geodesic hyperplanes not through the origin that reflect some given point $\mathbf{w} \in \mathbb{D}^n$ to the origin. This can be done through reflection in the hyperplane contained in the hypersphere with center $\mathbf{m} = \frac{\mathbf{w}}{||\mathbf{w}||^2}$ and radius $r = \sqrt{\frac{1}{||\mathbf{w}||^2} - 1}$. We easily verify that this hyperspherical inversion maps $\mathbf{w}$ to the origin.

$$R_{\mathbf{w}\to\mathbf{0}}(\mathbf{w}) = \frac{\mathbf{w}}{||\mathbf{w}||^2} + \frac{\frac{1}{||\mathbf{w}||^2} - 1}{||\mathbf{w} - \frac{\mathbf{w}}{||\mathbf{w}||^2}||}\left( \mathbf{w} - \frac{\mathbf{w}}{||\mathbf{w}||^2} \right) \tag{23}$$

$$= \frac{\mathbf{w}}{||\mathbf{w}||^2} + \frac{1 - ||\mathbf{w}||^2}{||\mathbf{w}||^4 - 2||\mathbf{w}||^2 + 1}\left( 1 - \frac{1}{||\mathbf{w}||^2} \right)\mathbf{w} \tag{24}$$

$$= \frac{\mathbf{w}}{||\mathbf{w}||^2} + \frac{1}{1 - ||\mathbf{w}||^2} \cdot \frac{||\mathbf{w}||^2 - 1}{||\mathbf{w}||^2}\mathbf{w} \tag{25}$$

$$= \frac{\mathbf{w}}{||\mathbf{w}||^2} - \frac{\mathbf{w}}{||\mathbf{w}||^2} = \mathbf{0}. \tag{26}$$

To show that this is a reflection in a geodesic hyperplane and, therefore, an isometry, we need to show that the hypersphere defined by $\mathbf{m}$ and $r$ is orthogonal to the boundary of $\mathbb{D}^n$. This is the case when all the triangles formed by the line segments between $\mathbf{0}$, $\mathbf{m}$ and any point $\mathbf{v}$ in the intersection of the hypersphere and the boundary of $\mathbb{D}^n$ are right triangles. This is exactly the case when the Pythagorean theorem holds for each of these triangles. For each $\mathbf{v}$ we have that $||\mathbf{v}|| = 1$ and $||\mathbf{v} - \mathbf{m}|| = r$, so

$$||\mathbf{v} - \mathbf{0}||^2 + ||\mathbf{v} - \mathbf{m}||^2 = 1 + r^2 \tag{27}$$

$$= \frac{1}{||\mathbf{w}||^2} \tag{28}$$

$$= \frac{||\mathbf{w}||^2}{||\mathbf{w}||^4} \tag{29}$$

$$= ||\mathbf{m} - \mathbf{0}||^2, \tag{30}$$

which shows that the Pythagorean theorem holds and, thus, that this hyperspherical inversion is a geodesic hyperplane reflection. The fact that such a hyperspherical inversion in a geodesic hyperplane is an isometry is shown in, for example, (Anderson, 2005).

## B. Evaluation metrics

We will use two distortion based evaluation metrics. The first one is the average relative distortion (Sala et al., 2018), given as

$$D_{ave}(\phi) = \frac{1}{N(N-1)} \sum_{u \neq v} \frac{|d_{\mathbb{D}}(\phi(u), \phi(v)) - d_T(u,v)|}{d_T(u,v)}, \tag{31}$$

where $N = |V|$ is the number of nodes. A low value for this metric is a necessary, but not sufficient condition for a high quality embedding, as it still allows for large local distortion. Therefore, we use a second distortion based metric, the worst-case distortion (Sarkar, 2011), given by

$$D_{wc}(\phi) = \max_{u \neq v} \frac{d_{\mathbb{D}}(\phi(u), \phi(v))}{d_T(u,v)} \left( \min_{u \neq v} \frac{d_{\mathbb{D}}(\phi(u), \phi(v))}{d_T(u,v)} \right)^{-1}. \tag{32}$$

$D_{ave}$ ranges from 0 to infinity and $D_{wc}$ ranges from 1 to infinity, with smaller values indicating strong embeddings. A large $D_{ave}$ indicates a generally poor embedding, while a large $D_{wc}$ indicates that at least some part of the tree is poorly embedded. Both values should be close to their minimum if an embedding is to be used for a downstream task. Lastly, another commonly used evaluation metric for unweighted trees is the mean average precision (Nickel & Kiela, 2017), given by

$$\text{MAP}(\phi) = \frac{1}{N} \sum_{u \in V} \frac{1}{\deg(u)} \sum_{v \in \mathcal{N}_V(u)} \frac{\left| \mathcal{N}_V(u) \cap \phi^{-1}\Big(B_{\mathbb{D}}(u,v)\Big) \right|}{\left| \phi^{-1}\Big(B_{\mathbb{D}}(u,v)\Big) \right|}, \tag{33}$$

where $\deg(u)$ denotes the degree of $u$ in $T$, $\mathcal{N}_V(u)$ denotes the nodes adjacent to $u$ in $V$ and where $B_{\mathbb{D}}(u,v) \subset \mathbb{D}^n$ denotes the closed ball centered at $\phi(u)$ with hyperbolic radius $d_{\mathbb{D}}(\phi(u), \phi(v))$, so which contains $v$ itself. The MAP reflects how well we can reconstruct neighborhoods of nodes while ignoring edge weights, making it less appropriate for various downstream tasks.

## C. Placing points on the vertices of a hypercube

The discussion here is heavily based on (Sala et al., 2018). We include it here for completeness. When placing a point on the vertex of an $n$-dimensional hypercube, there are $2^n$ options, so each option can be represented by a binary sequence of length $n$. For example, on a hypercube where each vertex $v$ has $||v||_\infty = 1$, each vertex is of the form $(\pm 1, \ldots, \pm 1)^T$, so we can represent $v$ as some binary sequence $s$. The distance between two such vertices can then be expressed in terms of the Hamming distance between the corresponding sequences as

$$d(v_1, v_2) = \sqrt{4 d_{\text{Hamming}}(s_1, s_2)},$$

which shows that points placed on vertices of a hypercube are maximally separated if this Hamming distance is maximized. This forms an interesting and well studied problem in coding theory where the objective is to find $k$ binary sequences of length $n$ which have maximal pairwise Hamming distances. There are some specific combinations of $n$ and $k$ for which optimal solutions are known, such as the Hadamard code. However, for most combinations of $n$ and $k$, the solution is still an open problem (MacWilliams & Sloane, 1977). Therefore, properly placing points on the vertices of a hypercube currently relies on the solution to an unsolved problem, making it difficult in practice.

## D. Proof of Theorem 3.1

*Proof.* For a tree $T = (V, E)$ with $N = |V|$, we know that the degrees of the vertices satisfy

$$\sum_{v \in V} \deg(v) = 2|E| = 2(N-1). \tag{34}$$

Suppose $W_1, \ldots, W_p \subset S^{n-1}$ are the sets of points on the hypersphere generated by the $p$ optimizations that need to be ran to perform the construction, then $|W_i| \neq |W_j|$, since we use the cached result whenever nodes have the same degree. Moreover, $|W_i|$ is equal to the degree of the node for which the points are generated, so

$$\sum_{i=1}^{p} |W_i| \leq \sum_{v \in V} \deg(v) = 2(N-1). \tag{35}$$

Given this constraint, the largest possible value of $p$ is when we can fit as many $|W_i|$'s in this sum as possible, which is when $|W_1|, \ldots, |W_p| = 1, \ldots, p$. In that case

$$\sum_{i=1}^{p} |W_i| = \sum_{i=1}^{p} i = \frac{p(p+1)}{2} \leq 2(N-1). \tag{36}$$

Solving for integer $p$ yields

$$p \leq \left\lceil \frac{1}{2}(\sqrt{16N - 15} - 1) \right\rceil. \tag{37}$$

$\square$

Note that this bound can be sharpened slightly by observing that each node $v$ with $\deg(v) > 1$ forces the existence of $\deg(v) - 1$ leaf nodes with degree 1. However, the asymptotic behaviour remains $\mathcal{O}(\sqrt{N})$.

## E. Proof of Theorem 4.2

*Theorem.* Given $\mathbf{x}, \mathbf{y} \in \mathbb{D}^n$ with $||\mathbf{x}|| < 1 - \epsilon^{t-1}$ and $||\mathbf{y}|| < 1 - \epsilon^{t-1}$, an approximation $d$ to equation 2 can be computed with FPE representations with $t$ terms and with a largest magnitude approximation to $\cosh^{-1}$ such that

$$\left| d - \cosh^{-1}\left( 1 + 2\frac{||\mathbf{x} - \mathbf{y}||^2}{(1 - ||\mathbf{x}||^2)(1 - ||\mathbf{y}||^2)} \right) \right| < \epsilon^*, \tag{38}$$

for some small $\epsilon^* > 0$.

*Proof.* We begin by noting that the accuracy of the largest magnitude approximation to $\cosh^{-1}$ depends on the underlying floating point algorithm used for computing the inverse hyperbolic cosine. While this function cannot be computed up to machine precision on its entire domain due to the large derivative near the lower end of its domain, it can still be computed quite accurately, i.e. there exists some small $\epsilon_1^* > 0$ such that

$$\left| \cosh^{-1}(x) - \cosh^{-1}(\tilde{x}) \right| < \epsilon_1^*, \tag{39}$$

where $x \in [1, R]$, where $R$ is the greatest representable number and $\tilde{x}$ is the floating point approximation to $x$, so for which we have

$$\frac{|\tilde{x} - x|}{|x|} < \epsilon. \tag{40}$$

For example, in PyTorch when using float64, we have $\epsilon_1^* \approx 2.107 * 10^{-8}$. If we can approximate the argument inside $\cosh^{-1}$ sufficiently accurately, then the largest magnitude approximation will be close enough to guarantee a small error. More specifically, let

$$z = 1 + 2\frac{||\mathbf{x} - \mathbf{y}||^2}{(1 - ||\mathbf{x}||^2)(1 - ||\mathbf{y}||^2)}, \tag{41}$$

and let $\tilde{z} = \tilde{z}_1 + \ldots + \tilde{z}_t$ with $|\tilde{z}_i| > |\tilde{z}_j|$ for each $i \neq j$ be the approximation to $z$ obtained through FPE arithmetic. If

$$\frac{|z - \tilde{z}|}{|z|} = \frac{|z - \sum_{i=1}^{t} \tilde{z}_i|}{|z|} < 2\epsilon, \tag{42}$$

where $\epsilon$ is the machine precision of the corresponding floating point format, then

$$|z - \tilde{z}_1| \leq |z - \tilde{z}| + \left| \sum_{i=2}^{t} \tilde{z}_i \right| \tag{43}$$

$$< 2\epsilon + 2\epsilon|\tilde{z}_1| \tag{44}$$

$$\leq 4\epsilon|z| + 2\epsilon|z - \tilde{z}_1|, \tag{45}$$

where we use that $|\tilde{z}_2| \leq \text{ulp}(\tilde{z}_1) = \epsilon|\tilde{z}_1|$, so that $|\sum_{i=2}^{t} \tilde{z}_i| < 2\epsilon|\tilde{z}_1|$. Now, we can rewrite to see that

$$\frac{|z - \tilde{z}_1|}{|z|} < \frac{4\epsilon}{1 - 2\epsilon} < 8\epsilon. \tag{46}$$

Therefore, by repeatedly using equation 39, we see that the largest magnitude approximation error is bounded by $16\epsilon_1^*$. Our ability to approximate the argument $z$ as precisely as in equation 42 using FPEs follows from the error bounds of the FPE arithmetic routines from (Popescu, 2017). This shows that the statement holds for $\epsilon^* = 16\epsilon_1^*$. $\qquad\square$

## F. Proof of Proposition 4.3

*Proposition.* The range of the inverse hyperbolic tangent formulation increases linearly in the number of terms $t$ of the FPEs being used.

*Proof.* When we use FPEs with $t$ terms, we can represent points $\mathbf{x}, \mathbf{y} \in \mathbb{D}^n$ such that $||\mathbf{x}|| = 1 - \epsilon^{t-1}$ and $||\mathbf{y}|| = 1 - \epsilon^{t-1}$. If we set $-\mathbf{y} = \mathbf{x} = (1 - \epsilon^{t-1}, 0, \ldots, 0)^T$, then

$$\cosh^{-1}\left(1 + 2\frac{||\mathbf{x} - \mathbf{y}||^2}{(1 - ||\mathbf{x}||^2)(1 - ||\mathbf{y}||^2)}\right) = \cosh^{-1}\left(1 + 4\frac{(1 - \epsilon^{t-1})^2}{(1 - (1 - \epsilon^{t-1})^2)^2}\right) \tag{47}$$

$$\geq \cosh^{-1}\left(1 + \frac{2}{4\epsilon^{2t-2} - 4\epsilon^{3t-3} + \epsilon^{4t-4}}\right) \tag{48}$$

$$\geq \cosh^{-1}\left(1 + \frac{2}{\epsilon^{2t-2}}\right) \tag{49}$$

$$= \log\left(1 + \frac{1}{2\epsilon^{2t-2}} + \sqrt{\left(1 + \frac{1}{2\epsilon^{2t-2}}\right)^2 - 1}\right) \tag{50}$$

$$\geq \log\left(\frac{1}{\epsilon^{t-1}}\right) \tag{51}$$

$$= (1 - t)\log(\epsilon) \tag{52}$$

$$= (t - 1)|\log(\epsilon)|, \tag{53}$$

which shows that we can compute a distance that is bounded from below by $\mathcal{O}(t)$. Similar steps can be used to show that the distance is also bounded from above by a $\mathcal{O}(t)$ term. $\qquad\square$

## G. Proof of Theorem 4.4

*Theorem.* Given a ulp-nonoverlapping FPE $x = \sum_{i=1}^{t} x_i \in [-1 + \epsilon^{t-1}, 1 - \epsilon^{t-1}]$ consisting of floating point numbers with a precision $b > t$, Algorithm 13 leads to an approximation $y$ of the inverse hyperbolic tangent of $x$ that satisfies

$$|y - \tanh^{-1}(x)| \leq \epsilon^*, \tag{54}$$

for some small $\epsilon^* > 0$.

*Proof.* The accuracy of the $x \in (-0.5, 0.5)$ branch of the algorithm follows easily from the accuracy of the algorithm for normal floating point numbers and the error bounds of the FPE routines from (Popescu, 2017), similar to the proof in Appendix E. The other branch can be a bit more problematic, due to the large derivatives near the boundary of the domain. For $0.5 \leq |x| < 1 - \epsilon^{t-1}$, we use

$$\tanh^{-1}(x) = 0.5 \cdot \text{sign}(x) \cdot \log\left(1 + \frac{2|x|}{1 - |x|}\right). \tag{55}$$

Let $z$ denote the argument of the logarithm, so

$$z = 1 + \frac{2|x|}{1 - |x|}, \tag{56}$$

and let $\tilde{z} = \tilde{z}_1 + \ldots + \tilde{z}_t$ denote that approximation of $z$ obtained through FPE operations. Due to the error bounds given in (Popescu, 2017), for FPEs with $t$ terms on the domain $0.5 \le |x| < 1 - \epsilon^{t-1}$ we can assume that

$$\frac{|z - \tilde{z}|}{|z|} < 2\epsilon, \tag{57}$$

where $\epsilon$ is the machine precision of the floating point terms. Now, since $|\tilde{z}_2| \le \text{ulp}(\tilde{z}_1) = \epsilon|\tilde{z}_1|$, we can write

$$|z - \tilde{z}_1| \le |z - \tilde{z}| + \Big|\sum_{i=2}^{t} \tilde{z}_i\Big| \tag{58}$$

$$\le 2\epsilon|z| + 2|\tilde{z}_2| \tag{59}$$

$$\le 2\epsilon|z| + 2\epsilon|\tilde{z}_1| \tag{60}$$

$$\le 4\epsilon|z| + 2\epsilon|\tilde{z}_1 - z|, \tag{61}$$

which can be rewritten as

$$|z - \tilde{z}_1| \le \frac{4\epsilon}{1 - 2\epsilon}|z| \le 8\epsilon|z|. \tag{62}$$

This shows that we can write $\tilde{z}_1 = (1 + \delta)z$, with $|\delta| < 8\epsilon$. Now, the error of the largest magnitude term approximation of the logarithm is

$$\Big|y - 0.5 \cdot \text{sign}(x) \cdot \log(z)\Big| = \Big|0.5 \cdot \text{sign}(\tilde{x}) \cdot \log(\tilde{z}_1) - 0.5 \cdot \text{sign}(x) \cdot \log(z)\Big| \tag{63}$$

$$= 0.5 \cdot \Big|\log\Big(\frac{z}{\tilde{z}}\Big)\Big| \tag{64}$$

$$= 0.5 \cdot \Big|\log\Big(\frac{\tilde{z}_1}{z}\Big)\Big| \tag{65}$$

$$= 0.5 \cdot \Big|\log\Big(\frac{(1 + \delta)z}{z}\Big)\Big| \tag{66}$$

$$= 0.5 \cdot |\log(1 + \delta)| \tag{67}$$

$$\le 0.5 \cdot |\delta| \tag{68}$$

$$\le 4\epsilon. \tag{69}$$

Lastly, we introduce some error through the approximation of the natural logarithm. However, as long as no overflow occurs, this error is typically bounded by the machine precision. Therefore, if we can approximate $z$ well enough, then we can guarantee an accurate computation of $\tanh^{-1}$. So combining this result with the error bounds from (Popescu, 2017) concludes the proof. $\qquad\square$

## H. Proof of Proposition 4.5

*Proposition.* The range of algorithm 13 increases linearly in the number of terms $t$.

*Proof.* The maximal values that we can encounter occur near the boundary of the domain, so set $x = 1 - \epsilon^{t-1}$. Then,

$$0.5 \cdot \text{sign}(x) \cdot \log\Big(1 + \frac{2|x|}{1 - |x|}\Big) = 0.5 \cdot \log\Big(1 + \frac{2 - 2\epsilon^{t-1}}{\epsilon^{t-1}}\Big) \tag{70}$$

$$\le 0.5 \cdot \log\Big(\frac{\epsilon^{t-1} + 2}{\epsilon^{t-1}}\Big) \tag{71}$$

$$\le 0.5 \cdot \log\Big(\frac{e}{\epsilon^{t-1}}\Big) \tag{72}$$

$$= 0.5 \cdot (1 - (t - 1)\log(\epsilon)) \tag{73}$$

$$= 0.5 \cdot (1 + (t - 1)|\log(\epsilon)|), \tag{74}$$

which shows that the range is bounded from above by $\mathcal{O}(t)$. A similar argument leads to a $\mathcal{O}(t)$ lower bound, showing that the range indeed increases linearly in the number of terms $t$. $\qquad\square$

## I. Binary tree embedding results for varying dimensions

Table 4 shows results of the embedding of a binary tree with float32 representations in 4, 7, 10 or 20 dimensions. Here, we have also tested an additional objective similar to MAM, where we use the cosines of the angles instead of the angles. We find that MAM generally leads to the best or close to the best results for each choice of dimensions.

*Table 4.* **Comparing hyperspherical separation** methods for the constructive hyperbolic embedding of a binary tree with a depth of 8 edges using float32 representations in 4, 7, 10 or 20 dimensions. ‡ uses Hadamard generated hypersphere points and ⋆ uses precomputed points from (Lovisolo & Da Silva, 2001).

| | $D_{ave}$ | | | | $D_{wc}$ | | | |
|---|---|---|---|---|---|---|---|---|
| | 4 | 7 | 10 | 20 | 4 | 7 | 10 | 20 |
| (Sala et al., 2018) ‡ | 0.734 | 0.734 | 0.734 | 0.734 | 1143 | 1143 | 1143 | 1143 |
| Sala et al. (2018) ⋆ | 0.235 | 0.502 | 0.361 | 0.726 | 10.51 | 132 | 18.42 | 280.5 |
| $E_0$ | 0.192 | **0.188** | 0.219 | 0.189 | 1.655 | 1.625 | 1.670 | 1.640 |
| $E_1$ | 0.190 | 0.196 | 0.204 | 0.190 | **1.619** | 1.664 | 1.686 | 1.698 |
| $E_2$ | 0.194 | 0.198 | 0.190 | 0.198 | 1.666 | 1.687 | 1.642 | 1.680 |
| Cosine similarity | 0.189 | 0.189 | **0.188** | **0.188** | 1.636 | 1.637 | **1.635** | 1.633 |
| MAM | **0.188** | **0.188** | **0.188** | 0.189 | 1.632 | **1.623** | **1.635** | **1.631** |

## J. Embedding $m$-ary trees in varying dimensions

Tables 5 and 6 show results of the embedding of various $m$-ary trees in dimensions 4, 7, 10 and 20, similar to Table 2. We find that MS-DTE gives the best results overall.

*Table 5.* **Comparison of average distortion of hyperbolic embedding algorithms on $m$-ary trees** with a maximum path length of $\ell = 8$. The h-MDS method is represented by †. The ‡ method is the combinatorial construction with the hyperspherical points being generated using the Hadamard construction, whereas the ⋆ method samples hyperspherical points from the precomputed points generated with the hyperspherical separation method from (Lovisolo & Da Silva, 2001). The h-MDS method outperforms the other methods for higher dimensions, but collapses nodes, making the embeddings unusable. HS-DTE has the best performance for smaller dimensions and second best performance for larger dimensions.

| $D_{ave}$ | 3-tree | | | | 5-tree | | | | 7-tree | | | |
|---|---|---|---|---|---|---|---|---|---|---|---|---|
| | 4 | 7 | 10 | 20 | 4 | 7 | 10 | 20 | 4 | 7 | 10 | 20 |
| (Sala et al., 2018) † | 0.09 | 0.07 | **0.03** | **0.01** | 0.18 | **0.05** | **0.04** | **0.03** | 0.16 | 0.13 | **0.03** | **0.02** |
| (Sala et al., 2018) ‡ | 0.11 | 0.11 | 0.11 | 0.11 | - | - | 0.12 | 0.12 | - | - | 0.12 | 0.12 |
| (Sala et al., 2018) ⋆ | 0.08 | 0.08 | 0.09 | 0.14 | 0.10 | 0.12 | 0.13 | 0.18 | 0.12 | 0.12 | 0.13 | 0.17 |
| HS-DTE | **0.06** | **0.06** | 0.06 | 0.06 | **0.09** | 0.09 | 0.09 | 0.09 | **0.10** | **0.10** | 0.10 | 0.10 |

## K. Embedding phylogenetic trees in varying dimensions

Additional experiments involving the phylogenetic trees with embedding dimensions 4, 7, 10 and 20 are shown in Tables 7, 8, 9 and 10. We observe that the precomputed points method struggles to separate points for higher dimensions, leading to higher distortion. Moreover, we find that HS-DTE gives the best results overall in every setting.

*Table 6.* **Comparison of worst-case distortion of hyperbolic embedding algorithms on $m$-ary trees** with a maximum path length of $\ell = 8$. The h-MDS method is represented by †. The ‡ method is the combinatorial construction with the hyperspherical points being generated using the Hadamard construction, whereas the ⋆ method samples hyperspherical points from the precomputed points generated with the hyperspherical separation method from (Lovisolo & Da Silva, 2001). HS-DTE has the best performance in all settings.

| $D_{wc}$ | 3-tree | | | | 5-tree | | | | 7-tree | | | |
|---|---|---|---|---|---|---|---|---|---|---|---|---|
| | 4 | 7 | 10 | 20 | 4 | 7 | 10 | 20 | 4 | 7 | 10 | 20 |
| (Sala et al., 2018) † | NaN | NaN | NaN | NaN | NaN | NaN | NaN | NaN | NaN | NaN | NaN | NaN |
| (Sala et al., 2018) ‡ | 1.14 | 1.14 | 1.14 | 1.14 | - | - | 1.14 | 1.14 | - | - | 1.14 | 1.14 |
| (Sala et al., 2018) ⋆ | 1.32 | 1.22 | 1.18 | 1.23 | 1.28 | 1.30 | 1.30 | 1.34 | 1.53 | 1.25 | 1.31 | 1.26 |
| HS-DTE | **1.07** | **1.07** | **1.07** | **1.07** | **1.14** | **1.10** | **1.10** | **1.10** | **1.14** | **1.13** | **1.12** | **1.12** |

*Table 7.* **Comparison of average distortion of hyperbolic embedding algorithms on the mosses and weevils trees.** ‡ represents the construction with Hadamard hyperspherical points and ⋆ the construction with points sampled from a set precomputed with (Lovisolo & Da Silva, 2001). The best performance is in bold. The embeddings are performed in a 4, 7, 10 or 20-dimensional space. Overall, we find that HS-DTE works best.

| $D_{ave}$ | Mosses | | | | Weevils | | | |
|---|---|---|---|---|---|---|---|---|
| | 4 | 7 | 10 | 20 | 4 | 7 | 10 | 20 |
| HypFPE + (Sala et al., 2018) ‡ | - | - | - | 0.09 | - | - | 0.07 | 0.07 |
| HypFPE + (Sala et al., 2018) ⋆ | 0.06 | 0.10 | 0.08 | 0.10 | **0.03** | 0.05 | 0.05 | 0.10 |
| HypFPE + HS-DTE | **0.04** | **0.04** | **0.04** | **0.04** | **0.03** | **0.03** | **0.03** | **0.03** |

*Table 8.* **Comparison of average distortion of hyperbolic embedding algorithms on the carnivora and lichen trees.** ‡ represents the construction with Hadamard hyperspherical points and ⋆ the construction with points sampled from a set precomputed with (Lovisolo & Da Silva, 2001). The best performance is in bold. The embeddings are performed in a 4, 7, 10 or 20-dimensional space. Overall, we find that HS-DTE works best.

| $D_{ave}$ | Carnivora | | | | Lichen | | | |
|---|---|---|---|---|---|---|---|---|
| | 4 | 7 | 10 | 20 | 4 | 7 | 10 | 20 |
| HypFPE + (Sala et al., 2018) ‡ | 0.04 | 0.04 | 0.04 | 0.04 | 0.12 | 0.12 | 0.12 | 0.12 |
| HypFPE + (Sala et al., 2018) ⋆ | **0.01** | 0.03 | **0.03** | 0.06 | 0.05 | 0.10 | 0.11 | 0.19 |
| HypFPE + HS-DTE | 0.02 | **0.02** | **0.03** | **0.02** | **0.06** | **0.06** | **0.05** | **0.05** |

## L. Statistics of the trees used in the experiments

Some statistics of the trees that are used in the experiments are shown in Table 11. Most notably, these statistics show that the true number of optimizations that has to be performed is significantly lower than the worst-case number of optimizations given by Theorem 3.1. To see this, note that an optimization step using MAM has to be performed each time a node is encountered with a degree that did not appear before. The result of this optimization step can then be cached and used for each node with the same degree.

## M. Graph and tree-like graph embedding results

The graphs that we test our method on are a graph detailing relations between diseases (Goh et al., 2007) and a graph describing PhD advisor-advisee relations (De Nooy et al., 2018). In order to embed graphs with the combinatorial constructions, the graphs need to be embedded into trees first. Following (Sala et al., 2018), we use (Abraham et al., 2007) for the graph-to-tree embedding. The results of the subsequent tree embeddings are shown in Table 12. These distortions are with respect to the tree metric of the embedded tree instead of with respect to the original graph. This is to avoid mixing the

*Table 9.* **Comparison of worst-case distortion of hyperbolic embedding algorithms on the mosses and weevils trees.** ‡ represents the construction with Hadamard hyperspherical points and ⋆ the construction with points sampled from a set precomputed with (Lovisolo & Da Silva, 2001). The best performance is in bold. The embeddings are performed in a 4, 7, 10 or 20-dimensional space. Overall, we find that HS-DTE works best.

| $D_{wc}$ | Mosses | | | | Weevils | | | |
|---|---|---|---|---|---|---|---|---|
| | 4 | 7 | 10 | 20 | 4 | 7 | 10 | 20 |
| HypFPE + (Sala et al., 2018) ‡ | - | - | - | 1.10 | - | - | 1.09 | 1.09 |
| HypFPE + (Sala et al., 2018) ⋆ | 1.36 | 1.21 | 1.14 | 1.16 | 1.25 | 1.12 | 1.11 | 1.13 |
| HypFPE + HS-DTE | **1.09** | **1.07** | **1.06** | **1.07** | **1.05** | **1.05** | **1.04** | **1.04** |

*Table 10.* **Comparison of worst-case distortion of hyperbolic embedding algorithms on the carnivora and lichen trees.** ‡ represents the construction with Hadamard hyperspherical points and ⋆ the construction with points sampled from a set precomputed with (Lovisolo & Da Silva, 2001). The best performance is in bold. The embeddings are performed in a 4, 7, 10 or 20-dimensional space. Overall, we find that HS-DTE works best.

| $D_{wc}$ | Carnivora | | | | Lichen | | | |
|---|---|---|---|---|---|---|---|---|
| | 4 | 7 | 10 | 20 | 4 | 7 | 10 | 20 |
| HypFPE + (Sala et al., 2018) ‡ | 6.76 | 6.76 | 6.76 | 6.76 | 43.4 | 43.4 | 43.4 | 43.4 |
| HypFPE + (Sala et al., 2018) ⋆ | 3.50 | 4.06 | 4.87 | 13.0 | 4.73 | 5.44 | 6.43 | 36.0 |
| HypFPE + HS-DTE | **2.46** | **2.45** | **2.03** | **2.35** | **4.07** | **4.63** | **3.30** | **7.17** |

influence of the tree-to-hyperbolic space embedding method with that of the graph-to-tree embedding.

From these results we again see that HypFPE + HS-DTE outperforms all other methods. However, it should be noted that graphs cannot generally be embedded with arbitrarily low distortion in hyperbolic space and that the graph to tree embedding method will introduce significant distortion. Hyperbolic space is not a suitable target for embedding a graph that is not tree-like. Therefore, we define our method as a tree embedding method and not as a graph embedding method.

## N. FPE arithmetic

---

**Algorithm 2** FPEAddition

---

1: **Input:** FPEs $x = x_1 + \ldots + x_n, y = y_1 + \ldots + y_m$ and number of output terms $r$.
2: $f \leftarrow \text{MergeFPEs}(x, y)$
3: $s \leftarrow \text{FPERenormalize}(f, r)$
4: **return** $s = s_1 + \ldots + s_r$

---

---

**Algorithm 3** MergeFPEs

---

1: **Input:** FPEs $x = x_1 + \ldots + x_n, y = y_1 + \ldots + y_m$.
2: $z \leftarrow \text{Concatenate}(x, y)$
3: Sort terms in $z$ in ascending order with respect to absolute value.
4: **return** Sorted $z = \{z_1, \ldots, z_{n+m}\}$.

---

**Algorithm 4** FPERenormalize

---

1: **Input:** List of floating point numbers $x = x_1, \ldots, x_n$ and number of output terms $r$.
2: $e \leftarrow \text{VecSum}(x)$
3: $y \leftarrow \text{VecSumErrBranch}(e, r)$
4: **return** $y = y_1 + \ldots + y_r$

---

**Algorithm 5** VecSum

---

1: **Input:** List of floating point numbers $x_1, \ldots, x_n$.
2: $s \leftarrow x_n$
3: **for** $i \in \{n - 1, \ldots, 1\}$ **do**
4: $\quad (s, e_{i+1}) \leftarrow 2\text{Sum}(x_i, s)$
5: **end for**
6: $e_1 \leftarrow s$
7: **return** $e_1, \ldots, e_n$

---

**Algorithm 6** VecSumErrBranch

---

1: **Input:** List of floating point numbers $e_1, \ldots, e_n$ and number of output terms $m$.
2: $j \leftarrow 1$
3: $\epsilon \leftarrow e_1$
4: **for** $i \in \{1, n - 1\}$ **do**
5: $\quad (r_j, \epsilon) \leftarrow 2\text{Sum}(\epsilon, e_{i+1})$
6: $\quad$ **if** $\epsilon \neq 0$ **then**
7: $\quad\quad$ **if** $j \geq m$ **then**
8: $\quad\quad\quad$ **return** $r_1, \ldots, r_m$
9: $\quad\quad$ **end if**
10: $\quad\quad j \leftarrow j + 1$
11: $\quad$ **else**
12: $\quad\quad \epsilon \leftarrow r_j$
13: $\quad$ **end if**
14: **end for**
15: **if** $\epsilon \neq 0$ **and** $j \leq m$ **then**
16: $\quad r_j \leftarrow \epsilon$
17: **end if**
18: **return** $r_0, \ldots, r_m$

---

**Algorithm 7** 2Sum

---

1: **Input:** floating point numbers $x$ and $y$.
2: $s \leftarrow \text{RN}(x + y)$ $\quad$ where RN is rounding to nearest
3: $x' \leftarrow \text{RN}(s - y)$
4: $y' \leftarrow \text{RN}(s - x')$
5: $\delta_x \leftarrow \text{RN}(x - x')$
6: $\delta_y \leftarrow \text{RN}(y - y')$
7: $e \leftarrow \text{RN}(\delta_x + \delta_y)$
8: **return** $(s, e)$

---

---

**Algorithm 8** Fast2Sum

---

1: **Input:** Floating point numbers $x$ and $y$ with $\lfloor \log_2 |x| \rfloor \geq \lfloor \log_2 |y| \rfloor$
2: $s \leftarrow \text{RN}(x + y)$
3: $z \leftarrow \text{RN}(s - x)$
4: $e \leftarrow \text{RN}(y - z)$
5: **return** $(s, e)$

---

**Algorithm 9** FPEMultiplication

---

1: **Input:** FPEs $x = x_1 + \ldots + x_n, y = y_1 + \ldots + y_m$, number of output terms $r$, bin size $b$ and precision $p$ (for float64: $b = 45, p = 53$).
2: $t_{x_1} \leftarrow \lfloor \log_2 |x_1| \rfloor$
3: $t_{y_1} \leftarrow \lfloor \log_2 |y_1| \rfloor$
4: $t \leftarrow t_{x_1} + t_{y_1}$
5: **for** $i \in \{1, \ldots, \lfloor r \cdot p/b \rfloor + 2\}$ **do**
6: $\quad B_i \leftarrow 1.5 \cdot 2^{t - ib + p - 1}$
7: **end for**
8: **for** $i \in \{1, \ldots, \min(n, r + 1)\}$ **do**
9: $\quad$ **for** $j \in \{1, \ldots, \min(m, r + 1 - i)\}$ **do**
10: $\quad\quad (\pi', e) \leftarrow 2\text{Prod}(x_i, y_j)$
11: $\quad\quad \ell \leftarrow t - t_{x_i} - t_{y_i}$
12: $\quad\quad sh \leftarrow \lfloor \ell/b \rfloor$
13: $\quad\quad \ell \leftarrow \ell - sh \cdot b$
14: $\quad\quad B \leftarrow \text{Accumulate}(\pi', e, B, sh, \ell)$
15: $\quad$ **end for**
16: $\quad$ **if** $j < m$ **then**
17: $\quad\quad \pi' \leftarrow x_i \cdot y_j$
18: $\quad\quad \ell \leftarrow t - t_{x_i} - t_{y_j}$
19: $\quad\quad sh \leftarrow \lfloor \ell/b \rfloor$
20: $\quad\quad \ell \leftarrow \ell - sh \cdot b$
21: $\quad\quad B \leftarrow \text{Accumulate}(\pi', 0, B, sh, \ell)$
22: $\quad$ **end if**
23: **end for**
24: **for** $i \in \{1, \ldots, \lfloor r \cdot p/b \rfloor + 2\}$ **do**
25: $\quad B_i \leftarrow B_i - 1.5 \cdot 2^{t - ib + p - 1}$
26: **end for**
27: $\pi \leftarrow \text{VecSumErrBranch}(B, r)$
28: **return** $\pi_1 + \ldots + \pi_r$

---

---

**Algorithm 10** Accumulate

---

1: **Input:** Floating point numbers $\pi', e$, list of floating point numbers $B$ and integers $sh, \ell$.
2: $c \leftarrow p - b - 1$
3: **if** $\ell < b - 2c - 1$ **then**
4:      $(B_{sh}, \pi') \leftarrow \text{Fast2Sum}(B_{sh}, \pi')$
5:      $B_{sh+1} \leftarrow B_{sh+1} + \pi'$
6:      $(B_{sh+1}, e) \leftarrow \text{Fast2Sum}(B_{sh+1}, e)$
7:      $B_{sh+2} \leftarrow B_{sh+2} + e$
8: **else if** $\ell < b - c$ **then**
9:      $(B_{sh}, \pi') \leftarrow \text{Fast2Sum}(B_{sh}, \pi')$
10:      $B_{sh+1} \leftarrow B_{sh+1} + \pi'$
11:      $(B_{sh+1}, e) \leftarrow \text{Fast2Sum}(B_{sh+1}, e)$
12:      $(B_{sh+2}, e) \leftarrow \text{Fast2Sum}(B_{sh+2}, e)$
13:      $B_{sh+3} \leftarrow B_{sh+3} + e$
14: **else**
15:      $(B_{sh}, p) \leftarrow \text{Fast2Sum}(B_{sh}, \pi')$
16:      $(B_{sh+1}, \pi') \leftarrow \text{Fast2Sum}(B_{sh+1}, \pi')$
17:      $B_{sh+2} \leftarrow B_{sh+2} + \pi'$
18:      $(B_{sh+2}, e) \leftarrow \text{Fast2Sum}(B_{sh+2}, e)$
19:      $B_{sh+3} \leftarrow B_{sh+3} + e$
20: **end if**
21: **return** $B$

---

**Algorithm 11** FPEReciprocal

---

1: **Input:** FPE $x = x_1 + \ldots + x_{2^k}$ an number of output terms $2^q$.
2: $r_1 = \text{RN}(\frac{1}{x_1})$
3: **for** $i \in \{1, \ldots, q\}$ **do**
4:      $v \leftarrow \text{FPEMultiplication}(r, x, 2^{i+1})$
5:      $w \leftarrow \text{FPERenormalize}(-v_1, \ldots, -v_{2^{i+1}}, 2.0, 2^{i+1})$
6:      $r \leftarrow \text{FPEMultiplication}(r, w, 2^{i+1})$
7: **end for**
8: **return** $r_1 + \ldots + r_{2^q}$

---

**Algorithm 12** FPEDivision

---

1: **Input:** FPEs $x = x_1 + \ldots + x_n, y = y_1 + \ldots + y_m$ and number of output terms $r$.
2: $z \leftarrow \text{FPEReciprocal}(y, m)$
3: $\pi \leftarrow \text{FPEMultiplication}(x, z, r)$
4: **return** $\pi$

---

**Algorithm 13** FPEtanh$^{-1}$

---

1: **Input:** FPE $\tilde{f} = \tilde{f}_1 + \ldots + \tilde{f}_t$.
2: **if** $|\tilde{f}| > 1$ **then**
3:      **return** NaN
4: **else if** $|\tilde{f}| = 1$ **then**
5:      **return** $\infty$
6: **else if** $|\tilde{f}| < 0.5$ **then**
7:      **return** $0.5 \cdot \text{sign}(\tilde{f}) \cdot \log(1 + 2|\tilde{f}| + \frac{2|\tilde{f}| \cdot |\tilde{f}|}{1 - |\tilde{f}|})$
8: **else**
9:      **return** $0.5 \cdot \text{sign}(\tilde{f}) \cdot \log(1 + \frac{2|\tilde{f}|}{1 - |\tilde{f}|})$
10: **end if**

*Table 11.* **Statistics for the trees used in the experiments.** The number of unique degrees is excluding nodes with a degree of 1. This number is equal to the total number of optimizations that has to be performed when embedding the tree using HS-DTE. The theoretical worst-case shows the worst-case number of optimizations that has to be performed according to Theorem 3.1. Note that the true number of optimizations is often significantly lower than this worst-case number.

| Tree | Nodes | Unique degrees | Theoretical worst-case | $\Delta(T)$ | Longest path length |
|---|---|---|---|---|---|
| $m$-ary trees | Varying | 2 | Varying | $m+1$ | varying |
| Mosses | 344 | 11 | 38 | 16 | 51 |
| Weevils | 195 | 5 | 29 | 8 | 29 |
| Carnivora | 548 | 3 | 45 | 4 | 192.4 |
| Lichen | 481 | 3 | 48 | 4 | 0.972 |

*Table 12.* **Comparison of hyperbolic embedding algorithms on graphs.** † represents the h-MDS method, ‡ the construction with Hadamard hyperspherical points and ⋆ the construction with points sampled from a set precomputed with (Lovisolo & Da Silva, 2001). The best float64 performance is underlined and the best FPE performance is in bold. All embeddings are performed in a 10-dimensional space. Hadamard generation cannot be used, since each embedded graph has a $\Delta(T)$ greater than 8. HypFPE + HS-DTE outperforms all methods.

| | Precision | Diseases | | CS PhDs | |
|---|---|---|---|---|---|
| | | $D_{ave}$ | $D_{wc}$ | $D_{ave}$ | $D_{wc}$ |
| (Nickel & Kiela, 2017) | 53 | 0.40 | NaN | 0.72 | NaN |
| (Ganea et al., 2018) | 53 | 0.85 | 4831 | 0.94 | 803 |
| (Yu et al., 2022b) | 53 | 0.72 | 1014 | 0.91 | 1220 |
| (Sala et al., 2018) † | 53 | 0.06 | NaN | 0.08 | NaN |
| (Sala et al., 2018) ‡ | 53 | - | - | - | - |
| (Sala et al., 2018) ⋆ | 53 | 0.364 | 5.07 | 0.33 | 3.84 |
| HS-DTE | 53 | 0.28 | 2.28 | 0.29 | 2.76 |
| HypFPE + (Sala et al., 2018) ‡ | 417 | - | - | - | - |
| HypFPE + (Sala et al., 2018) ⋆ | 417 | 0.05 | 1.16 | **0.04** | 1.14 |
| HypFPE + HS-DTE | 417 | **0.04** | **1.14** | **0.04** | **1.09** |

