# OpenReview forum: "Low-distortion and GPU-compatible Tree Embeddings in Hyperbolic Space"
_ICML.cc/2025/Conference — ICML 2025 poster_

### Official Review · Reviewer_uxsR · 2025-02-15

**Overall Recommendation:** 3

**Summary:**

The Authors propose a pipeline to embed trees in hyperbolic space with minimal distortion yet with the ability to operate on accelerated GPU by the generalised Dalaunay embedding algorithm with minimal angle maximization and providing a solid framework to essentially increase the precision by using multiple float data even when the precision of each variable is limited.

## update after rebuttal.

The authors have provided acceptable evidence that embedding pure trees can contribute to the ICML community, e.g., through action search. It should be included in the manuscript, but it can be easily done in the camera-ready version. I have raised the score to Weak Accept.

**Claims And Evidence:**

The goal of the paper (low-distortion with fixed length float numbers, exploiting hyperbolic space with a general dimensionality) itself is simple, and the proposed method is designed to directly solve the issue. Algorithm 1 is a natural generalisation of Sarker's embedding in high-dimensional hyperbolic space. Applying the minimal angle maximization is a natural and direct idea to put the children of a node uniformly. Using multiple fixed precision float numbers also directly solves the issue of the lack of precision required by hyperbolic embedding. They are both convincing.

**Essential References Not Discussed:**

Nothing.

**Experimental Designs Or Analyses:**

The experiments are designed soundly so that they can directly evaluate how well the proposed method achieved the initial goal of the paper.

**Methods And Evaluation Criteria:**

As discussed above, the proposed method succeeds in solving existing issues of hyperbolic embedding of a tree.

**Other Comments Or Suggestions:**

Considering the solid contribution of the Authors' work to computer science, I humbly suggest the following:
- The most straightforward way is to submit the manuscript to another venue (it requires withdrawal from ICML), which can appreciate the Authors' contribution better. In any case, I encourage the Authors to upload the work to some online repositories to secure the priority right if they have not done so.
- If the Authors want to present their work on machine learning venues, rewrite it from scratch so that the contribution to the community is clear.
In any case, the competitors would be data structures, like the distance matrix, naive implementation of a tree as a graph, etc., rather than existing hyperbolic representations.

**Other Strengths And Weaknesses:**

As a computer science paper, the paper's contribution is solid and almost complete. Having said that, regrettably, I need to flag that the current manuscript of the paper does not clarify the contribution to the ICML community, where the main focus is on machine learning, as its name suggests. Many hyperbolic representation learning methods, represented by (Nickel and Kiela 2017), the motivation was in learning some representations of a graph, typically not a tree strictly but approximated by a tree, and those obtained representations could be used for link prediction. This is a procedure of retrieving some initially unknown information from data, which can be called "learning." However, the proposed method only considers where **the complete information of a pure tree** is available. Where we have the complete information of a pure tree and obtain perfect representations in hyperbolic space that can recover the original distance structure, what do we "learn" by that? In other words, why are we not satisfied with the original tree? Actually, I do not see any information that we can obtain from the Authors' method, compared to the original tree. Hence, I do not dare to call the Authors' method "machine learning." Recall that (Sarker 2011) was presented in the graph drawing context, and (Sala et al., 2018) included hyperbolic MDS, which has room for being called "machine learning."

Obviously, even if some work is not directly related to machine learning, provided that it has the potential to contribute to the machine learning community, ICML must be capable of accepting it as long as the potential is clarified in the paper. However, the Authors' current manuscript does not clarify how it can contribute to the community.

In fact, the Authors' work has the potential to contribute to the machine-learning community. The perfect representations can provide the distance between two nodes at almost constant time (though it depends on the precision of float and dimension, strictly speaking) while the space complexity is almost linear to the number of nodes (again, it depends on the precision of float and dimension). This is in contrast to the naive distance matrix implementation needing computational costs quadratic to the number of nodes. This must be an attractive *data structure*, if not machine learning itself, for machine learning on tree data. Nevertheless, the current manuscript does not stress such perspectives. It requires rewriting from scratch, which is not what the rebuttal period aims at.

**Questions For Authors:**

Could you formalize the statements of Theorems 4.2. and 4.4?

**Relation To Broader Scientific Literature:**

The paper might have more contributions to the data compression or data structure contexts.

**Theoretical Claims:**

The statement of Theorem 4.2. must be rewritten. "Some $\\epsilon^*$" makes no sense mathematically. If some argued $\\epsilon^*=10^{300}$ is still small, there would be no practical implication of Theorem 4.2. If it were a function of $\\epsilon$, $t$, or machine epsilon, it might have some practical implication.
Perhaps, the Authors wanted to say that we can let $\\epsilon^*$ be 16 times the machine epsilon, according to the proof? Theorem 4.4. has the same issue.

---

> ### Author Rebuttal · Authors · 2025-03-31
>
> We thank the reviewer for their kind remarks regarding the contribution of our paper and their recognition of the convincing nature of our method. Below we address the main concern of the reviewer, namely the potential and relevance to the machine learning community.
>
> **Relevance to the machine learning community.**
> In our view, there are three important reasons why this paper should be published in a machine learning conference such as ICML.
>
> 1. This paper follows a long tradition of hyperbolic embedding papers published in machine learning conferences. As mentioned by the reviewer, (Sala et al., 2018) is closest to our work and was previously published in ICML. Their h-MDS performs an eigenvalue decomposition on the pairwise distance matrix and projects the resulting eigenvectors to the hyperboloid to obtain embeddings, hence also not making use of any learning algorithms. The high impact of (Sala et al., 2018) serves as an indication of the potential of hyperbolic tree embeddings within our field. Other hyperbolic embedding papers in machine learning conferences include (Nickel and Kiela, 2017; Ganea et al., 2018; Yu et al., 2022b), each of which uses a learning approach to obtain embeddings. Although these methods all employ some type of learning, they achieve considerably worse results than the constructive approaches. Therefore, we think it is important that our constructive approach is published within the machine learning community.
>
> 2. We believe that the primary relevance of hyperbolic embeddings to the machine learning community lies in their potential for downstream deep learning applications. In recent years, a common approach for incorporating hierarchical knowledge into deep learning models is by placing the hyperbolic embeddings of the hierarchies on top of neural networks. Then, the node embeddings serve as prototypes for learning. See for example (Liu et al., 2020; Long et al., 2020; Yu et al., 2022b). In other words, hyperbolic embeddings are already an active part of the deep learning pipeline. Lower distortion means better embeddings which leads to better results in hierarchical deep learning. Hence it is important for us to share our findings in a machine learning venue to maximize impact.
>
> 3. We believe FPEs to be an important direction for the field of hyperbolic machine learning. Nearly every paper on hyperbolic learning mentions the numerical problems that arise from attempting to do computation in hyperbolic space, with a particularly compelling analysis of the numerical stability by [1], also published at ICML. They show that the maximally representable subset of hyperbolic space in the commonly used models is severely limited in its radius. FPEs alleviate these problems, significantly increasing this radius. We think our work will be impactful to hyperbolic machine learning by shining a light on the potential of FPEs while also providing an implementation of the current state-of-the-art routines on this type of arithmetic that is compatible with the most commonly used deep learning framework, i.e., PyTorch. This provides a starting point for exciting future work on hyperbolic machine learning with higher precision, while mantaining GPU-compatibility.
>
> Based on these, we believe our work will be most impactful if it is published at ICML where it can reach its intended target audience: the hyperbolic machine learning community. We hope that the reviewer will appreciate and agree with our motivations. We will make sure to better clarify these important points in the paper.
>
> [1] Mishne, Gal, et al. "The numerical stability of hyperbolic representation learning." ICML. 2023.
>
> **Corrections to Theorems 4.2 and 4.4.**
> The reviewer is correct in that we intend $\epsilon^*$ to be in the order of the machine epsilon. We apologize for the lack of clarity and have changed the text accordingly.

---

> > ### Comment · Reviewer_uxsR · 2025-04-05
> >
> > Thank you for your comments. Unfortunately, I do not think your reply and references directly answer my concern because they are not directly related to what you are doing: converting a pure tree to points in hyperbolic space where no cycle is allowed. Simply, how can converting a pure tree (without any cycle allowed) to points in hyperbolic space contribute to the ICML community? Please distinguish a tree-**like** structure and a **pure** tree since your method only applies to a pure tree.

---

> > > ### Author Response · Authors · 2025-04-07
> > >
> > > Hyperbolic embeddings of pure trees have been actively used in deep learning, making it of high relevance for the ICML community. A common approach is to use the pure tree as prior knowledge for a down-stream task and use the hyperbolic embedding of the tree as prototypes on top of any neural network. For example, (Yu et al., 2022b) use the hyperbolic embedding of a hierarchical classification system of skin lesions, given as a pure tree, to improve skin lesion recognition. (Long et al., 2020) and (Ghadimi Atigh et al., 2021) both use the embedding of a hierarchy of actions, also given as a pure tree, in hyperbolic space to improve action recognition. Other examples include [1], which uses the embedding a taxonomy of butterflies in the form of a pure tree on top of a neural network, or [2], which uses hyperbolic tree embeddings to improve explainability in action recognition.
> > >
> > > To clarify, our method does apply to tree-like structures as well, as shown in the experiments in Appendix M, where we have applied our method to the Diseases and CS PhDs graphs. To use our method on such structures, we combine HS-DTE with some method for embedding a tree-like graph into a tree such as (Abraham et al., 2007). This is the same approach that (Sala et al., 2018) take to apply their construction to tree-like structures. Therefore, methods that use the hyperbolic embedding of tree-like structures can also benefit from our method. There are several examples of such papers, such as [3], which uses a continuous graph embedding to improve aggregation across nodes of the embedded graph in GCNs. (Liu et al., 2020) uses a hyperbolic embedding of the WordNet noun hierarchy to improve zero-shot recognition. Another example can be found in [4], which uses a hyperbolic graph embedding to improve zero-shot learning.
> > >
> > > We would also like to point out that our theory and implementations for FPEs form a basis for building GPU-compatible neural networks with higher precision in general. The potential of such networks is significant, particularly in hyperbolic machine learning, where nearly every method suffers from the numerical problems that originate from a lack of precision and which are extensively analyzed in [5]. Therefore, we believe our paper to be relevant to the machine learning community not only due to the improved tree and tree-like graph embeddings, but also because it paves the road for higher precision neural networks.
> > >
> > > Following the guidance of the reviewer, we will update the paper to highlight the importance of tree embeddings in hyperbolic space for machine learning.
> > >
> > > [1] Dhall, Ankit, et al. "Hierarchical image classification using entailment cone embeddings." CVPR workshops, 2020.
> > >
> > > [2] Gulshad, Sadaf, Teng Long, and Nanne van Noord. "Hierarchical explanations for video action recognition." CVPR, 2023.
> > >
> > > [3] Pei, Hongbin, et al. "Geom-gcn: Geometric graph convolutional networks." ICLR, 2020.
> > >
> > > [4] Xu, Yan, et al. "Meta hyperbolic networks for zero-shot learning." Neurocomputing 491 (2022): 57-66.
> > >
> > > [5] Mishne, Gal, et al. "The numerical stability of hyperbolic representation learning." ICML, 2023.

---

### Official Review · Reviewer_gaL1 · 2025-02-20

**Overall Recommendation:** 4

**Summary:**

Existing combinatorial approaches to embedding trees in hyperbolic space suffer from issues stemming from (1) the difficulty of spreading out points on a hypersphere, and (2) floating-point precision issues. To address issue (1), the authors propose highly-separated Delauney tree embeddings (HS-DTE), which maximize the minimal angle between two leaves, unlike previous approaches which optimize the mean pairwise angles. To address (2), the authors present HypFPE, a modernized approach to floating point expansion with special attention paid to the hyperbolic distance function. The authors demonstrate the superiority of their methods against comparable approaches for a variety of real and empirical trees.

## Update after rebuttal
I maintain my recommendation to accept this paper. A common theme among other reviews is that, while the paper itself is of high quality, its relevance to the ICML readership is limited. I would like to offer a dissenting opinion: as the authors themselves argue better than I could in their rebuttal to Reviewer uxsR, there is substantial precedent for machine learning venues publishing work pertaining to tree embeddings. More speculatively, I also believe it is valuable to expose ML venue audiences to more nuts-and-bolts hyperbolic work, given the extensive history of cross-pollination between these fields.

**Claims And Evidence:**

The authors do an excellent job situating their work within the rich literature on embedding trees in hyperbolic space, motivating their approach in terms of empirical issues and theoretical problems (the Tammes problem), proving bounds on all of their formulas, and devising careful experiments that demonstrate their improvement over other methods. All in all, each part of their paper is clearly isolated, well-argued, and relevant to the central argument.

**Essential References Not Discussed:**

The authors perform extensive benchmarking on phylogenetic trees, but neglect to cite any of the literature concerning embedding phylogenetic trees in hyperbolic space [1, 2, 3, 4]. These tend to be much more Bayesian in flavor, as is typical of the phylogenetics literature more broadly; I acknowledge that this would be difficult to do explicit comparisons with, but I think acknowledging the existence of this literature is necessary.

**References**

[1] Macaulay et al (2023). Fidelity of hyperbolic space for Bayesian phylogenetic inference. https://journals.plos.org/ploscompbiol/article?id=10.1371/journal.pcbi.1011084.

[2] Jian et al (2022). Learning Hyperbolic Embedding for Phylogenetic Tree Placement and Updates. https://pmc.ncbi.nlm.nih.gov/articles/PMC9495508/.

[3] Matsumoto et al (2021). Novel metric for hyperbolic phylogenetic tree embeddings. https://pmc.ncbi.nlm.nih.gov/articles/PMC8058397/.

[4] Wilson (2021). Learning phylogenetic trees as hyperbolic point configurations. https://arxiv.org/pdf/2104.11430.

[5] Chen et al (2025). Variational Combinatorial Sequential Monte Carlo for Bayesian Phylogenetics in Hyperbolic Space. https://arxiv.org/abs/2501.17965.

**Experimental Designs Or Analyses:**

I closely read the description of the experiments and Appendix B and verified they are reasonable.

**Methods And Evaluation Criteria:**

The authors devise fair and minimal evaluations for demonstrating the value of their method. Their extension to phylogenetic trees is also reasonable. Given the emphasis on GPU optimization and several allusions to runtime improvements throughout the paper, I am confused by the absence of a runtime analysis in the paper.

**Other Comments Or Suggestions:**

* The discussion of hyperbolic reflections is somewhat confusing, and I found the transition into reflections in subsection 2.1 quite abrupt. For context, I am quite familiar with the preceding math, and not at all familiar with reflections. To this end, I have three concrete suggestions:
  * Add a paragraph break prior to starting this discussion
  * Explaning the motivation for the reflections more clearly would motivate this math better.
  * A figure demonstrating a hyperbolic reflection in $\mathbb{P}^2$ could be helpful
* Per the ICML style guide, table captions should appear above the tables.
* It is unclear what the horizontal lines in Tables 2 and 3 mean
* The running title of the paper still reads "Submission and Formatting Instructions for ICML 2025"

**Other Strengths And Weaknesses:**

**Strengths:**
* The paper is very coherently written, and the experiments are minimal and convincing.
* The authors promise to provide Pytorch-compatible libraries for both HS-DTE and HypFPE, which will be a boon to the hyperbolic deep learning community.

**Weaknesses:**
* The novelty of this paper is somewhat limited: as I understand it, HS-DTE is a simple substitution of one loss function for another relative to the previous Delauney embeddings work; similarly, floating point expansion has been done in hyperbolic space previously, and the authors simply modernize it and give a more careful gloss of its role in the hyperbolic distance function. That said, this is a well-studied problem, so I expect most papers in this field to be somewhat incremental. Given the clear improvements the authors demonstrate, I am less concerned about this weakness.
* The phylogenetics benchmarks are not situated in the context of other hyperbolic methods for embedding/inferring phylogenetic trees.
* The authors talk about runtime, but do not provide actual benchmarks for this. Memory usage would also be interesting to look at, especially as a function of the number of FPE terms.

**Questions For Authors:**

* Can you explain the "elbow" in Figure 1(b)? Is this surprising, or is it explained by some theoretical aspect of FPE?
* I found it very surprising that it is a difficult problem to distribute $n$ points nicely on $\mathbb{S}^d$. Can you provide an intuitive explanation for why this is the case?

**Relation To Broader Scientific Literature:**

The authors situate themselves in the context of tree embedding literature, which they subdivide into optimization-based and constructive methods; this method belongs to the latter, but quite reasonably the authors benchmark against all methods of both categories. They discuss the relationship of HS-DTE to Delauney tree embeddings, as well as explain how their work on HypFPE differs from other work on floating-point expansion. All in all, the authors are quite clear about where their work fits in to the existing literature.

**Theoretical Claims:**

I skimmed the proofs in Appendices D–H, but I did not scrutinize them closely.

---

> ### Author Rebuttal · Authors · 2025-03-31
>
> We thank the reviewer for their positive feedback regarding the writing, motivation and results. Below we address the points and questions raised by the reviewer.
>
> **Runtime comparison and memory usage.**
> Given the similar theoretical complexities, we consider benchmarking this to be an interesting and important direction for future work. However, currently, such a runtime comparison cannot be performed fairly, since (Sala et al., 2018) perform arbitrary precision (AP) arithmetic in Julia, which uses the heavily optimized GNU MPA and GNU MPFR libraries, written in C, which are the result of well over two decades of development. Our FPE library is a first version Python library, that still requires extensive optimization. So, before we can make a fair comparison, an optimized CUDA version of our library should first be developed. We expect that, once such a library exists, the runtime of AP arithmetic versus FPE arithmetic will be similar for small tensors, while for large tensors the FPE arithmetic will likely be significantly faster due to its ability to leverage the benefits of accelerated hardware.
>
> Regarding the memory usage, the main difference between FPE arithmetic and AP arithmetic is that each term of an FPE contains its own exponent (11 bits per float64), whereas an AP floating point number has a single exponent (64 bits in Julia). Therefore, adding bits of precision is slightly less efficient for FPEs than for AP arithmetic. However, this difference is marginal.
>
> **Reference to literature on the embedding of phylogenetic trees.**
> We thank the reviewer for pointing out the papers on the embedding of phylogenetic trees. We have added references to these papers in the manuscript.
>
> **Clarification on hyperbolic reflections.**
> We thank the reviewer for their suggestions for further clarification on hyperbolic reflections. Each of these will be incorporated into the manuscript, including a figure to show some simple examples of such reflections in $\mathbb{D}^2$. The motivation for reflections lies within the fact that these are isometries of hyperbolic space, which means that we can safely use these to move our embeddings around without changing the distortion. This fact is not trivial and is a result that is sometimes proved in books on hyperbolic geometry such as in (Anderson, 2005). A reference to this result should have been in the manuscript and we have since added this to Appendix A alongside a clear explanation of the motivation and interpretation.
>
> **Explanation behind elbow behaviour in Figure 1(b).**
> This is interesting behaviour that results from increasing the scaling factor $\tau$ within the construction itself. Note that, as we increase the precision, we increase the scaling factor $\tau$, since a greater $\tau$ is always beneficial as long as it does not lead to numerical problems. For smaller values of $\tau$, the method cannot make good use of the curvature and, as a result, subtrees will tend to overlap as we iteratively embed a tree, leading to massive distortion. As we increase the scaling factor $\tau$, there comes a point where the subtrees stop overlapping, causing the worst-case distortion to quickly drop to near 1. Past this threshold, increasing $\tau$ will only marginally decrease distortion compared to the initial drop, as the distortion will already be very low. We will include this explanation to the paper.
>
> **Intuitive explanation behind the difficulty of the Tammes problem.**
> This is a very surprising problem as the problem statement itself is deceptively simple. Giving an intuitive explanation of why exactly this problem is so difficult is not particularly easy itself. Probably, the most straightforward way of getting a feeling for the difficulty of this problem is by manually attempting to place points on the sphere $S^2$. Placing a small number of points remains fairly simple (for example through the use of regular polyhedra), but as the number of points increases, this quickly becomes very difficult. In fact, for 15 points, there is already no known optimal solution [1]. This problem has sparked enormous amounts of research and has been found to have applications in many different areas of physics, computer science, and other scientific fields. For further information we refer the reviewer to a book on spherical coding such as [2].
>
> [1] D. A. Kottwitz, The densest packing of equal circles on a sphere, Acta Cryst. Sect. A 47 (1991), 158–165.
>
> [2] Conway, John Horton, and Neil James Alexander Sloane. Sphere packings, lattices and groups. Vol. 290. Springer Science and Business Media, 2013.
>
> **Other comments and suggestions.**
> We have incorporated these suggestions into the manuscript. Thank you.
>
> We hope to have adequately addressed the reviewers concerns and to have answered their questions.

---

> > ### Comment · Reviewer_gaL1 · 2025-04-01
> >
> > Thank you for your thoughtful answers to my review. I have no further questions, and am keeping my score at a 4.

---

### Official Review · Reviewer_xLwr · 2025-03-14

**Overall Recommendation:** 2

**Summary:**

This paper takes on the task of transforming into an algorithm some mathematical results about low-distortion embeddability of any metric tree into hyperbolic space, whose existence and non-quantitative construction was known by work of Sarkar.

This endeavor leads to two new challenges.
(1) the tree embeddings can become very large (unbounded diameter). Since the Poincare disk model sends the infinite hyperbolic space to a bounded chart, this means that unbounded numerical precision has to be used. Thus the authors present encodings via floating-point formal sums, that fit well with GPU processing and allow arbitrary precision
(2) at vertices at which the metric tree has high degree N, for good efficiency (i.e. to get low distortion) we would impose that the N branches have to go in as well separated as possible directions, as measured in the tangent space to hyperbolic space at that point.. thus since in the tangent space is euclidean, we get the problem of distributing N points on a sphere, as uniformly as possible. This is in itself a well studied problem with many possible versions, but (a) one version has to be chosen and (b) it needs to be fit within the framework of hyperbolic space embeddings of trees. Here for (a) the authors choose to implement an energy depending on angle separations, and they proceed to do (b) in the natural way one would expect.

**Claims And Evidence:**

I think that the claims about the algorithm working better for tree embeddings is warranted, both by the theoretical and by the quantitative comparison with other embeddings, so I don't have much to say on that.

However I don't follow the reasoning about this embedding being useful for deep learning frameworks, as hinted at by the passages in the introduction in the first page of the paper.
(In fact the authors mention this limitation explicitly in lines 247-248, first column, but they don't propose any solution.)
So I don't see how the floating point expansion from Def. 4.1 can be easily fit into a neural network processing. If the whole goal is just to embed a known tree into hyperbolic space, OK. But in practice for HNN applications one does not know the correct tree, and that has to be learned by a neural network or some other mechanism. The compatibility of the floating point expansion with Deep Learning frameworks has not been discussed nor tested. So the claims about usefulness in machine learning remain dubious to me.

**Essential References Not Discussed:**

I think that some reference to the book of Borodachov-Hardin-Saff would be useful.
Also work by Carlos Beltran and collaborators on energy minimization over spheres can be interesting to mention. https://arxiv.org/abs/1703.00416. Perhaps taking points randomly from a determinantal point process can be another way to build good enough embeddings for other choices of energies?

**Experimental Designs Or Analyses:**

I think that the analyses match closely the theoretical part, and are not the most important part of the paper.
However, they are sound for me.

**Methods And Evaluation Criteria:**

I think that yes, they do.

**Other Comments Or Suggestions:**

Line 133-134 second column: "strong embeddings" means what precisely? do you mean less distorted? And "numerical issues" means precisely what? It'd be good to be specific about what you refer to.

Line 136 second column: I think that "deg_{max}" would be better as "maxdeg", if the underlying language of reference is English. If you follow French or Italian then go with "deg_max" but that seems odd.

Line 145 second column: "should be \tau= \Omega(1)" what do you mean by "should" exactly? should be that in order to do what?

Line 154 firlst column "However" means what, i.e. what's the problem with the methodology of that library? Be precise please.

In step 5 of Algorithm 1, what is R_{\phi(v)\to 0} ? this notation is not introduced before.

Line 189-190 second column: "We find that this method results in strong separation when compared to highly specialized existing methods [..]" can you explain how you measured that? What's the comparison and to what do you compare it?

In the statement of Thm. 3.1 you talk about "number of optimizations" what does that mean? how is that defined from eq. (9) exactly?

At the end of page 4, you say "MAM is easily optimizable", what does that mean? Does it have no local minima? Or what does it mean "easily" and what would "hardly" mean?

Line 223 first column: "can be further optimized", so why was it not optimized? maybe write "could be optimized"?

About the description of formula (13), in the preceding paragraph: "unevaluated sums" does that mean "collections of terms" or what does it mean exactly? maybe represent that as tuples of floating point numbers, and then use formula (13) just for the interpretation. This is because "unevaluated sums" seems odd and to me unnecessarily hard to imagine.

**Other Strengths And Weaknesses:**

I covered all the important points in other sections.

**Questions For Authors:**

1) Do you know of actual separation guarantees that allow to compare the theoretical results of separation obtained from your algorithm for mininmizing (13) vs the previous works? In other words, in theory, what's the improvement, how can that be measured?

2) I reiterate on the question of NN processing compatibility of your floating point expansions.

**Relation To Broader Scientific Literature:**

As said before, I think that this paper relates as a practical counterpart to the mathematical theory of embedding trees into hyperbolic space, however it does not connect with the most influential part of Hyperbolic Machine Learning, due to the presence of floating point expansions and due to not proposing a way to include this into hyperbolic machine learning pipelines.

**Theoretical Claims:**

Yes, I checked all the details and didn't find anything wrong.

---

> ### Author Rebuttal · Authors · 2025-03-31
>
> We thank the reviewer for their helpful and constructive feedback. Below, we describe how the feedback has been incorporated into the paper and address the reviewer's concerns.
>
> **The importance of embedding given hierarchies in deep learning.**
> Many works have shown the strong potential of deep learning with hyperbolic embeddings. All these works assume that hierarchical information is known *a priori*. Such knowledge is important for deep learning and can be incorporated into the model (as prototypes for example), the loss function, or several other means. The most intuitive examples of such hierarchies can be found in biology, where tasks such as classification can greatly benefit from knowledge regarding the evolution of species, typically given through animal taxonomies or phylogenetic trees [1, 2]. Other examples can be found in many different areas, such as biochemistry [3], medicine (Yu et al., 2022b), action recognition (Long et al., 2020), commonly used classification datasets such as ImageNet [4] and many more. In general, these works show that hierarchical information benefits deep learning, especially for hierarchical classification. There are various ways in which such a hierarchy can be leveraged to improve performance, such as through prototype learning.
>
> Our embedding approach can be directly used in deep learning in the same way. For all deep learning approaches, the better the embedding, the better the corresponding prototype distribution, benefitting deep learning. This is shown in (Liu et al., 2020; Long et al., 2020; Yu et al., 2022b). We apologize for not making this case clear enough and we will update the discussion in the introduction accordingly.
>
> ~
>
> [1] Stevens, Samuel, et al. "Bioclip: A vision foundation model for the tree of life." CVPR. 2024.
>
> [2] Chen, Jingzhou, et al. "Label relation graphs enhanced hierarchical residual network for hierarchical multi-granularity classification." CVPR. 2022.
>
> [3] Yazdani-Jahromi, Mehdi, et al. "HELM: Hierarchical Encoding for mRNA Language Modeling." ICLR. 2025.
>
> [4] Chatterjee, Ankita, Jayanta Mukherjee, and Partha Pratim Das. "ImageNet classification using wordnet hierarchy." IEEE TAI 5.4 (2023): 1718-1727.
>
>
> **Fitting floating point expansions into neural networks.**
> Integrating floating point expansions into existing deep learning libraries is an exciting next step, although it is indeed not completely trivial. Our statement in lines 247-248 is about the incompatibility of arbitrary precision arithmetic, since this incompatibility is at the hardware level. On the other hand, FPE arithmetic does allow for the use of GPUs, but requires new routines for additional functions and, often, for their derivatives in order to work with the usual deep learning pipelines. More specifically, to incorporate FPE arithmetic into existing hyperbolic learning methodology, we would have to implement the forward and backward passes of each operation involving the hyperbolic space (where the added precision is warranted) in FPE arithmetic, similar to what we did with the computation of the distance function in this work. On top of the additional required theory, the new architectures that incorporate these embeddings would have to be designed, optimized and tested. Because of this, we consider this to be outside the scope of the current paper, but a very interesting and necessary direction that we intend to explore in future work.
>
> **Additional references.**
> We thank the reviewer for the additional references and have added these to the paper.
>
> **Theoretical comparison between our method and existing methods in terms of separation.**
> There are a few specific settings in which optimal solutions are known. In some of those cases we have compared the optimal result to ours and found very similar separation. However, ours has the benefit of being always applicable, whereas the table of best known solutions (Cohn, 2024) has many missing entries. Combining both could lead to a small improvement, but due to the marginal increase in separation between our results and the best known results, the decrease in distortion will also be marginal.
>
> **Other comments and suggestions.**
> We have incorporated each of the reviewer's comments and suggestions into the text. Due to the character limit we cannot address each point individually here. If the reviewer would like further clarification on any point in particular, then we would be happy to address these in the next response.
>
> We hope to have addressed the reviewer's concerns and that they are convinced of the potential of our method for the hyperbolic learning community.

---

> > ### Comment · Reviewer_xLwr · 2025-04-01
> >
> > I have read the other reviews and their rebuttals, and the rebuttal to my review.
> >
> > - I liked a lot the review of gaL1, which included nice references on phylogenetic trees, some of which I dindn't know and want to read soon. I agree that phylogenetic tree analysis is a good direction of application for the current work, although I don't think that  this would be necessarily done by ML methods. I didn't find their questioning on novelty addressed in the rebuttal to their review, but agree with them that it is minor.
> >
> > - About the review of uxsR, I share the admiration for the implementation side and I consider it too a strong paper in general, and I have exactly the same concern about not suitability to the ICML community, since the paper does not work in a ML framework and requires exciting new work if we want to plug it into a ML pipeline. I feel that similarly to my review's rebuttal, their rebuttal does not address the concern accurately.
> >
> > This is in consonance with my points A. and B. below, which highlight points not addressed in the rebuttal. My view is that the paper in which point B. will be covered will have strong impact in conferences such as ICML, but at the moment we are not yet there.
> >
> > ----------------About my own review's rebuttal: ------------
> >
> > A. About "All these works assume that hierarchical information is known a priori."
> >
> > Sorry, I strongly disagree. The whole point of most classical HNNs is to find out a good candidate for the hierarchy, and often the goal is just to process data with a hidden hierarchy in a way that respects that structure, without necessarily embedding it explicitly.
> >
> > Most works that use the hyperbolic metric that I know, use the metric precisely because the user knows that there is some hierarchical structure, but this structure is not known a priori.
> >
> > One of the main benefits of hyperbolic space is that it gives a continuous search space as a substitute to the combinatiorial space of all metric trees. If we were to search for the metric tree that best fits a given distance matrix, the search space faces a combinatiorial explosion, as the number of non isomorphic combinatorial trees with N leaves grows exponentially in N. Instead, hyperbolic embeddings plug the N leaves in hyperbolic space with low deformation and optimizes this data in the natural geometry, sometimes selecting a good candidate for almost-optimal embedding, but without fixing the hierarchy a priori.
> >
> > Your work fixes the hierarchy a priori and then embeds it into hyperbolic space. This does not allow to circumvent the combinatorial explosion of the space of trees mentioned above. This is why it is unfair to compare your work to the ones in which the hierarchy is not assumed known a priori.
> >
> > A few examples about the hierarchy not being known a priori:
> >
> > 1) the "Hyperbolic neural network" paper by Ganea et al. uses as experiments some random perturbation of strings. Then they validate the hyperbolic embedding's tendency of selecting good hierarchical relations.
> > 2) "Poincare embeddings for learning hierarchical representations" -- the title says it already: hierarchical structures are learned, not known/fixed and then embedded.
> > 3) "Hyperbolic entailment cones for learning hierarchical embeddings" -- the title says it, as before.
> > 4) "From Trees to Continuous Embeddings and Back: Hyperbolic Hierarchical Clustering" -- the goal is to use hyperbolic space as a continuous search space for tree structures. This means that the task does not assume a known tree structure.
> > 5) "Hyperbolic Busemann Learning with Ideal Prototypes" -- again, the setting is an 1 or 2, with the same comment
> >
> > Note that all these except for 1) are taken from the beginning of your reference list, they are classical papers on hyperbolic neural networks.
> >
> > B. I agree that adapting floating point expansion to deep learning in actual computations is exciting and outside the scope, thanks for the discussion.
> >
> > C. When I asked for theoretical comparison on separation I meant provable results that describe the comparison at a theoretical level Empirical validations are OK but there are many ingredients in these that can vary and modify the outcome, therefore I don't find them equally compelling.  Can you please either provide information about the theoretical comparison, or affirm that this is outside the scope of the paper?
> >
> > I thank the authors for their responses and for the moment I'll keep my score as is, since I don't see my above concerns A. and B. addressed, similarly to those of reviewer uxsR mentioned below.

---

> > > ### Author Response · Authors · 2025-04-02
> > >
> > > We are glad to hear that the reviewer agrees on point B regarding the application of FPEs to deep learning. Aside from their application to deep learning, FPEs can be used to improve hyperbolic representations in all current applications, so we do consider this a relevant contribution nonetheless. For point C, we would also be very interested in such an analysis. Unfortunately, the analyses go beyond the current understanding of the Tammes problem in mathematics, hence the theoretical comparisons are currently not viable and we have to fall back on empirical comparisons.
> > >
> > > **This leaves point A as the remaining  point. We agree that not all hyperbolic learning papers assume the a priori availability of hierarchical knowledge. However, it is a common starting point in a lot of machine learning papers, including multiple of the papers mentioned by the reviewer. Out of the 5 papers listed by the reviewers 3 actually do assume hierarchies as prior knowledge. This is a crucial point.**
> > >
> > > Our goal is to embed a (symbolic) tree-like structure into a continuous hyperbolic embedding space. The Poincaré Embeddings (NeurIPS) and Hyperbolic Entailment Cones (ICML) papers have the exact same goals as our method and are also baselines in our paper:
> > > 1. "Poincaré embeddings for learning hierarchical representations": In this paper, some set of nouns is embedded in hyperbolic space. The loss that is used (equation 6 of the paper) uses the set $\mathcal{D}$, which is the set of hypernymy relations between these noun pairs. In other words, the set of nouns are the vertices and the set $\mathcal{D}$ the edges between these vertices. Thus, the loss explicitly assumes a given graph that is directly used to embed the nodes of the graph into hyperbolic space. So, yes, this paper does assume a given hierarchy.
> > > 2. "Hyperbolic entailment cones for learning hierarchical embeddings": Same as for Poincaré embeddings, the loss function (equation 32 in the paper) explicitly uses a hierarchy that is known *a priori* in the form of hypernym links. This is, again, simply a set of edges on some set of nouns, which together form a graph. This graph is then sampled for nouns that are directly related and ones that are not (sets $P$ and $N$ in the equation).
> > >
> > > Other papers with this exact same setup for embedding a given tree-like structure include (Sala et al., ICML 2018) and (Yu et al., MICCAI 2022).
> > >
> > > Second, there is a wealth of papers that tackle a down-stream deep learning task with a hyperbolic embedding space on top of a neural network. Many of these papers assume that for their classification task, a hierarchy is available a priori that spans all classes. The hierarchy is then incorporated into the model through various methods such as embedding the hierarchy into hyperbolic space and using these as prototypes. Examples include, but are not limited to, (Liu et al., 2020; Long et al., 2020; Ghadimi Atigh et al., 2021; Ghadimi Atigh et al., 2022; Yu et al., 2022b). It even includes the Hyperbolic Busemann Learning paper mentioned by the reviewer, which includes settings where hierarchical prior information can be injected, see for example Tables 5 and 6 of that paper.
> > >
> > > The reviewer is right that not all papers assume hierarchies, e.g., Hyperbolic Neural Networks focus on re-interpreting core neural network layers in hyperbolic space and are not focused on hierarchical embeddings. Papers such as [1] try to infer hierarchies from hyperbolic embeddings.
> > >
> > > **In conclusion, our setup is a standard setup in hyperbolic machine learning literature.** We show that in this setup, we provide the best solution and the best path forward in hyperbolic learning with known hierarchies. It would be heartbreaking for us if the paper is rejected on this premise, since it is simply not true that hyperbolic learning papers never assume hierarchies. We hope to have convinced the reviewer of the validity of our setup and the value of our contribution.
> > >
> > > [1] Nickel, Maximillian, and Douwe Kiela. "Learning continuous hierarchies in the Lorentz model of hyperbolic geometry." ICML. PMLR, 2018.

---

### Official Review · Reviewer_bqVV · 2025-03-14

**Overall Recommendation:** 3

**Summary:**

This paper proposes a construction based tree embedding method in hyperbolic space.

**Claims And Evidence:**

"While these approaches are flexible due to minimal assumptions, the optimization can be unstable, slow and result in heavily distorted embeddings"

This is not true, many hyperbolic embeddings achive great results, e.g., in  NLP, knowledge graphs, GNNs. It would be great to show more evidences for the claim "...result in heavily distorted embeddings".

**Essential References Not Discussed:**

NA

**Experimental Designs Or Analyses:**

The experimental designs on synthetic datasets make senses, but should also be evaluated on real tree-like datasets. Otherwise, it is not convincing to claim that construction based methods are better than learning based methods.

**Methods And Evaluation Criteria:**

This paper focus on tree embeddings, and only evaluate the model on small and synthetic tree data.

To demonstrate its applications to real-world cases, it should also be evaluated on real tree-like (but not eactly tree) datasets, such as on tasks of hierarchical graph (WordNet) embeddings, knowledge graphs, etc. Another reasoning to consider real data is to compare with learning based hyperbolic embeddings used in KG embeddings, GNNs, etc.

**Other Comments Or Suggestions:**

See Weaknesses

**Other Strengths And Weaknesses:**

Strengths:

1) I like the paper, the writing of the paper is great. Motivation is also clear.
2) The theorems provide some nice guarantees that are useful for real applciations (and the other methods do not have).
3) Ilustration of the results and findings (figures and tables) are also clear.

Weaknesses:

1) Comparision w.r.t efficiency is missing
2) Lack of evaluation on real-world datasets and ML tasks
3) Lack of conceptual description of the proposed idea. I suggest to add a figure 1 in the intro, to summerize the main idea of the paper.

**Questions For Authors:**

Does the proposed method only work for trees? or it can be extended to DAG, or any tree-like data like hierarchies but with a few cycles?

**Relation To Broader Scientific Literature:**

This work might be related to many applications involving hierarchical data.

**Theoretical Claims:**

I read the theorem, but not check the proof.

---

> ### Author Rebuttal · Authors · 2025-03-31
>
> We thank the reviewer for their positive feedback on the writing and clarity, and we are glad to hear that they like the paper. The reviewer points out four points of discussion and suggestions: the performance of optimization-based methods, experimental results on real-world data, comparison w.r.t. efficiency and a figure 1 for summarizing the approach. We address each of these below.
>
> **Performance of optimization-based methods.**
> The approaches we mention regarding distorted embeddings are about the optimization-based approaches Poincaré Embeddings and Hyperbolic Entailment Cones specifically. We will clarify our statement to avoid any confusion with hyperbolic embeddings in general. There are several ways of measuring the quality of embeddings. For some of these, such as the MAP and F1 score, the optimization-based methods indeed perform well. This is in line with their intended downstream tasks which often involve link prediction. However, on distance-based metrics, these methods tend to perform significantly worse, as shown for instance by (Sala et al., 2018) and by our results. On the other hand, the constructive based approaches perform well on both types of metrics, while being significantly faster and without requiring hyperparameter tuning. We will add the point on quality with respect to metrics to the discussion.
>
> **Experimental results on real-world data and tree-like data.**
> We fully agree with the reviewer, which is why we have included results on the embedding of real-world phylogenetic trees in Subsection 5.3 and tree-like graphs in Appendix M of the manuscript. Even on such real-world data, we find that our method has the strongest performance. Our method can be used to embed graphs in the same manner employed by (Sala et al., 2018). We hope the reviewer will find these results as convincing as we do.
>
> **Efficiency comparison.**
> We believe comparisons can be made at two levels. The first is a comparison in efficiency between HS-DTE and the constructive approaches by (Sala et al., 2018). Here, the difference in efficiency comes from the difference in generating points on the hypersphere in step 6 of Algorithm 1. This generation for our method takes mere seconds and, according to Theorem 3.1, only has to be performed $\mathcal{O}(\sqrt{N})$ times in the worst-case (much less in practice). So, in our experiments we find a minimal increase (only seconds) in computation time with respect to (Sala et al., 2018).
>
> The second is the efficiency of floating point expansion arithmetic compared to arbitrary precision arithmetic. Here, we have referred to the theoretical guarantees described for instance in (Popescu, 2017), which tell us that the complexities between the two types of arithmetic are similar. Testing this in practice is an important future direction. Currently, this cannot be done fairly as we would be comparing our first version Python implementation of FPE arithmetic against the heavily optimized arbitrary precision arithmetic that is used by Julia. More specifically, Julia uses the heavily optimized GNU MPA and GNU MPFR libraries, written in C, which have been in development for well over two decades with the sole purpose of building highly optimized and easily applicable arbitrary precision arithmetic. To compare fairly against such libraries, an optimized CUDA version of our library should be developed. We believe this is an exciting future research direction.
>
> We expect that an optimized version for FPE arithmetic will significantly outperform arbitrary precision arithmetic libraries, especially for large tensors, as our approach can rely on accelerated hardware. We believe more attention should be brought to FPEs through papers such as this, as that will hopefuly help speed up the development of such libraries. We will include the discussion on efficiency and open research opportunities to the paper.
>
> **Addition of a figure 1 to summarize the method.**
> We agree with reviewer and thank them for the suggestion. We have added a Figure 1 to the manuscript.
>
> We thank the reviewer for their helpful feedback and hope that we have addressed their concerns.

---

### Decision · Program_Chairs · 2025-05-01

**Decision:**

Accept (poster)

**Comment:**

The submitted manuscript introduces a novel tree embedding method in hyperbolic space. After reading the initial reviews, the rebuttal and following the discussion, my summary is that all reviewers agree that the paper presents an interesting novel idea and is technically strong. Maybe some claims and assertions should be toned down a bit, especially regarding the a-priori use of hierarchical information. Also, the authors provided a convincing rebuttal, clarifying most concerns raised by the reviewers. From that perspective, I think there are no substantial changes to the manuscript required and all concerns can be addressed in a camera-ready version without an additional round of review. Hence, I am recommending acceptance. Please incorporate all (reasonable) changes requested when preparing the final manuscript.